



# Improvement from the satellite-derived NO_X emissions on air quality modeling and its effect on ozone and secondary inorganic aerosol formation in Yangtze River Delta, China

Yang Yang[1], Yu Zhao[1,2*], Lei Zhang[1], Jie Zhang[3], Xin Huang[4], Xuefen Zhao[1],

Yan Zhang[1], Mengxiao Xi[1] and Yi Lu[1]

1. State Key Laboratory of Pollution Control & Resource Reuse and School of the Environment, Nanjing University, 163 Xianlin Ave., Nanjing, Jiangsu 210023, China

2. Jiangsu Collaborative Innovation Center of Atmospheric Environment and Equipment Technology (CICAEET), Nanjing University of Information Science & Technology, Jiangsu 210044, China

3. Jiangsu Provincial Academy of Environmental Science, 176 North Jiangdong Rd., Nanjing, Jiangsu 210036, China

4. School of the Atmospheric Sciences, Nanjing University, 163 Xianlin Ave., Nanjing, Jiangsu 210023, China

*Corresponding author: Yu Zhao

Phone: 86-25-89680650; email: yuzhao@nju.edu.cn





**Abstract**

We developed a "top-down" methodology combining the inversed chemistry
transport modeling and satellite-derived tropospheric vertical column of $NO_2$, and
estimated the $NO_X$ emissions of Yangtze River Delta (YRD) region at a horizontal
resolution of 9 km for January, April, July and October 2016. The effect of the
top-down emission estimation on air quality modeling, and the response of ambient
ozone ($O_3$) and secondary inorganic aerosols ($SO_4^{2-}$, $NO_3^-$, and $NH_4^+$, SNA) to the
changed precursor emissions were evaluated with the Community Multi-scale Air
Quality (CMAQ) system. The top-down estimates of $NO_X$ emissions were smaller
than those in a national emission inventory, MEIC (i.e., the "bottom-up" estimates),
for all the four months, and the monthly mean was calculated at 260.0 Gg/month,
24% less than the bottom-up one. The $NO_2$ concentrations simulated with the
bottom-up estimate of $NO_X$ emissions were clearly higher than the ground
observation, indicating the possible overestimation in current emission inventory
attributed to its insufficient consideration of recent emission control in the region. The
model performance based on top-down estimate was much better, and the biggest
change was found for July with the normalized mean bias (NMB) and normalized
mean error (NME) reduced from 111% to -0.4% and from 111% to 33%, respectively.
The results demonstrate the improvement of $NO_X$ emission estimation with the
nonlinear inversed modeling and satellite observation constraint. With the smaller
$NO_X$ emissions in the top-down estimate than the bottom-up one, the elevated
concentrations of ambient $O_3$ were simulated for most YRD and they were closer to
observation except for July, implying the VOC (volatile organic compound)-limit
regime of $O_3$ formation. With available ground observations of SNA in the YRD,
moreover, better model performance of $NO_3^-$ and $NH_4^+$ were achieved for most
seasons, implying the effectiveness of precursor emission estimation on the
simulation of secondary inorganic aerosols. Through the sensitivity analysis of $O_3$
formation for April 2016, the decreased $O_3$ concentrations were found for most YRD
region when only VOCs emissions were reduced or the reduced rate of VOCs



emissions was two times of that of $NO_X$, implying the crucial role of VOCs control on
$O_3$ pollution abatement. The SNA level for January 2016 was simulated to decline
12% when 30% of $NH_3$ emissions were reduced, while the change was much smaller
with the same reduced rate for $SO_2$ or $NO_X$. The result suggests that reducing $NH_3$
emissions was the most effective way to alleviate SNA pollution for YRD in winter.

## 1.   Introduction

Nitrogen oxides ($NO_X = NO_2 + NO$) play an important role on the formation of

ambient ozone ($O_3$) and secondary inorganic aerosol (SIA). The $NO_X$ emission
inventories are necessary input of the air quality model (AQM), and have a great
influence on the simulation particularly for $NO_2$, $O_3$ and SIA (Zhou et al., 2017; Chen
et al., 2019a). Moreover, it is crucial for exploring the sources of atmospheric
pollution of $O_3$ and fine particles (particles with aerodynamic diameter smaller than
$2.5\,\mu m$, $PM_{2.5}$) with AQM.

The $NO_X$ emission inventories were usually developed with a bottom-up method,

in which the emissions were calculated based on the activity data (e.g., fuel
consumption and industrial production) and emission factors (the emissions per unit
of activity data) by source category and region. Bias existed commonly in the
bottom-up inventories, due mainly to the uncertainty of economic and energy
statistics and fast changes in the emission control measures, especially in developing
countries like China (Granier et al., 2011; Saikawa et al., 2017; Zhang et al., 2019). To
improve the emission estimation, an inversed "top-down" method has been developed
based on satellite observation and AQM (Martin et al., 2003; Zhao and Wang et al.,
2009; Zyrichidou et al., 2015; Yang et al., 2019a). The emissions were corrected
based on the difference between the modeled and observed tropospheric vertical
column densities (TVCDs) of $NO_2$, and the response coefficient of $NO_2$ TVCDs to
emissions ((Martin et al., 2003; Cooper et al., 2017). With higher temporal and spatial
resolution than other instruments, the $NO_2$ TVCDs from Ozone Monitoring
Instrument (OMI) were frequently used (Kurokawa et al., 2009; Gu et al., 2014; de





Foy et al., 2015; Kong et al., 2019; Yang et al., 2019a).
Currently, the top-down methods were mainly developed at the global or national
scale with relatively coarse horizontal resolution (Martin et al., 2003; Miyazaki et al.,
2012; Jena et al., 2014). For example, Martin et al. (2003) and Miyazaki et al. (2012)
estimated the global top-down $NO_X$ emissions at the horizontal resolution of $2°×2.5°$
and $2.8°×2.8°$, respectively. As reported by Martin et al. (2003), the satellite-derived
$NO_X$ emissions for 1996-1997 were higher than bottom-up ones by 50-100% in the Po
Valley, Tehran, and Riyadh urban areas. Miyazaki et al. (2012) suggested that the
$NO_X$ emissions were underestimated with the bottom-up method over eastern China,
eastern United States, southern Africa, and central-western Europe. In India, the
top-down estimation of annual $NO_X$ emission at the horizontal resolution of $0.5°×$
$0.5°$ was 7-60% smaller than various bottom-up ones in 2005 (Jena et al., 2014). With
the TVCDs from OMI and another instrument (Global Ozone Monitoring Experiment,
GOME), the difference in $NO_X$ emission estimation for China was quantified at 0.4
Tg N/yr at the resolution of $70×70$ km (Gu et al., 2014). The estimates were limited at
the regional scale with finer resolution. In China, great differences exist in the levels
and patterns of air pollution across the regions, attributed partly to a big variety of air
pollutant sources across the country. To achieve the target of air quality improvement
required by the central government, varied air pollution control plans were usually
developed and implemented at the city/provincial levels. Therefore, the top-down
estimates in $NO_X$ emissions at finer horizontal resolution are in great need for
understanding the primary sources of $NO_2$ pollution and demonstrating the effect of
emission control at the regional scale.
Biases existed in the top-down estimates resulting from the uncertainties of the
inversed method and satellite observation (Cooper et al., 2017; Ding et al., 2017; Liu
et al., 2019; Yang et al., 2019a; b), and they could further influence the reliability of
AQM and the rationality of control measures. At present, those estimates of $NO_X$
emissions were usually evaluated with satellite observation. For example, the bias
between the $NO_2$ TVCDs from OMI observation and AQM based on the top-down
$NO_X$ emission estimation was $-30.8 ± 69.6 × 10^{13}$ molecules $cm^{-2}$ in winter in India





(Jena et al., 2014). The linear correlation coefficient ($R^2$) between OMI and AQM
with the top-down emission estimates could reach 0.84 in Europe (Visser et al., 2019).
Compared to the satellite observation with relatively large uncertainty (Yang et al.,
2019b; Liu et al., 2019), surface concentrations that better represent the effect of air
pollution on human health and the ecosystems were less applied in the evaluation of
the top-down estimates of $NO_X$ emissions. Limited studies were conducted at coarse
horizontal resolutions at the national scale. For example, Liu et al. (2018) found that
the normalized mean error (NME) between the observed and simulated $NO_2$
concentrations based on the top-down estimate of $NO_X$ emissions could reach 32% in
China at the resolution of $0.25° \times 0.25°$. Besides $NO_2$, the estimation of $NO_X$
emissions also play an important and complicated role on simulation of secondary air
pollutant concentrations including $O_3$ and SIA, and the response of secondary
pollution to the primary emissions was commonly nonlinear. For example, Wang et al.
(2019) found that the simulated $O_3$ concentrations in Shanghai (the most developed
city in eastern China) could increase over 20% with a 60% reduction in $NO_X$
emissions in summer 2016, implying a clear "VOC-limit" pattern for the $O_3$ formation
in the mega city. For the response of SIA to $NO_X$ emissions, the $NH_4^+$ and $SO_4^{2-}$
concentrations at an urban site in another mega city Nanjing in eastern China were
simulated to increase 1.9% and 2.8% with a 40% abatement of $NO_X$ emissions in
autumn 2014, respectively, due to the weakened competition of SIA formation against
$SO_2$ (Zhao et al., 2020). To our knowledge, however, the relatively new information
from the inversed modeling of $NO_X$ emissions has not been sufficiently incorporated
into the SIA and $O_3$ analyses with AQM in China.

Located in eastern China, the Yangtze River Delta (YRD) region including the

city of Shanghai and the provinces of Anhui, Jiangsu and Zhejiang is one of the most
developed and heavy-polluted regions in the country. The air quality for most cities in
YRD failed to meet National Ambient Air Quality Standard (NAAQS) Class II in
2016 (MEPPRC, 2017). $NO_X$ emissions made great contributions to the severe air
pollution in the region. Based on an offline-sampling and measurement study, for
example, the annual average of the $NO_3^-$ mass fraction to the total $PM_{2.5}$ reached 19%



in Shanghai in 2014, and it was significantly elevated in the pollution event periods
(Ming et al., 2017). In this study, we chose the YRD to estimate the $NO_X$ emissions
with the inversed method and to explore their influence on the air quality modeling.
The top-down estimates in $NO_X$ emissions were firstly obtained with the nonlinear
inversed method and OMI-derived $NO_2$ TVCDs for 2016. The advantage of the
top-down estimation against on the bottom-up one was then evaluated with the AQM
and abundant ground-based $NO_2$ concentrations. The influences of the top-down
estimation in $NO_X$ emissions were further detected on $O_3$ and SIA modeling.
Sensitivity analyses were conducted by changing the emissions of precursors to
investigate the sources and potential control approaches of $O_3$ and SIA pollutions for
the region.

## 152     2. Data and Methods

### 153     2.1 The top-down estimation of $NO_X$ emissions

The top-down estimation of $NO_X$ emissions was conducted for January, April,
July, and October of 2016, representing the situations of the four seasons in the YRD
region, and the horizontal resolution was $9 \times 9$ km. The inversed method assumed a
nonlinear and variable correlation between $NO_X$ emissions and $NO_2$ TVCDs (Cooper
et al., 2017), and the a posterior daily emissions (top-down estimates) were calculated
with the following equations:
$$E_t = E_a \left( 1 + \frac{\Omega_o - \Omega_a}{\Omega_o} \beta \right) \tag{1}$$

$$\frac{\Delta E}{E} = \beta \frac{\Delta \Omega}{\Omega} \tag{2}$$

where $E_t$ and $E_a$ represent the a posterior and the a prior daily $NO_X$ emissions,
respectively; $\Omega_o$ and $\Omega_a$ represent the observed and simulated $NO_2$ TVCDs,
respectively; $\beta$ represents the response coefficient of the simulated $NO_2$ TVCDs to a
specific change in emissions, and was calculated based on the simulated changes in
TVCDs ($\Delta \Omega$) from a 10% changes in emissions ($\Delta E$). For a given month, the a
posterior daily emissions were used as the a priori emissions of the next day, and the



monthly top-down estimate of the NO$_X$ emissions was scaled from the average of the
a posterior daily emissions of the last three days in the month, as the top-down
estimate of daily NO$_X$ emissions usually converged within a one-month simulation
period (Zhao and Wang, 2009; Yang et al., 2019b).

The NO$_2$ TVCDs were from OMI onboard the Aura satellite. It crosses the

equator at 1:30 PM of local time. The horizontal resolution of OMI was 24 $\times$ 13 km at
nadir (Levelt et al., 2006), one of the finest resolutions available for NO$_2$ TVCD
observation before October 2017. We applied the Peking University Ozone
Monitoring Instrument NO$_2$ product (POMINO v1, Lin et al., 2014; Lin et al., 2015)
to constrain the NO$_X$ emissions. POMINO v1 modified the retrieval methodology of
the Dutch Ozone Monitoring Instrument NO$_2$ product (DOMINO v2) in China, and
provided better linear correlation of NO$_2$ TVCDs between the satellite and available
ground-based observations with the multi-axis differential optical absorption
spectroscopy (MAX-DOAS) (Lin et al., 2015). The original NO$_2$ TVCDs from
POMINO v1 (level 2) were resampled into an 18$\times$18 km grid system based on the
area weight method, and then downscaled to 9$\times$9 km with the Kriging interpolation.
As an example, the NO$_2$ TVCDs for July 2016 in the YRD are shown in Figure S1 in
the supplement, and larger TVCDs were found in the east-central YRD.

**2.2 Model configuration**

The Models-3 Community Multi-scale Air Quality (CMAQ) version 5.1 was

used to conduct the inversed modeling of NO$_X$ emission estimation and to simulate
the ground-level concentrations of NO$_2$, O$_3$ and SIA. As a three-dimensional Eulerian
model, CMAQ includes complex interactions of atmospheric chemistry and physics
and is one of the most widely applied AQM to evaluate the sources and processes of
air pollution in China (UNC, 2012; Xing et al., 2015; Zheng et al., 2017). As shown in
Figure 1, the two nested modeling domains were applied with their horizontal
resolutions set 27 and 9 km, respectively. The mother domain (D1, 177$\times$127 cells)
included most parts of China, and the second (D2, 118 $\times$ 121 cells) covered the YRD
region. The model included 28 vertical layers and the height of the first layer (ground





layer) was approximately 60 m. The carbon bond gas-phase mechanism (CB05) and
AERO6 aerosol module were used in the CMAQ. The initial concentrations and
boundary conditions for the D1 were derived from the default clean profile, while
those of D2 were extracted from the CMAQ Chemistry Transport Model (CCTM)
outputs of its mother domain. The first 5 days of each simulated month were chosen
as the spin-up period. Details on model configuration were described in Zhou et al.
(2017) and Yang and Zhao (2019).
The Multi Resolution Emission Inventory for China (MEIC,
http://www.meicmodel.org/) was applied as the initial input of anthropogenic
emissions in D1 and D2, with an original horizontal resolution at $0.1°{\times}0.1°$. In this
study, the MEIC emissions from residential source were downscaled to the horizontal
resolution of $9{\times}9$ km based on the spatial density of population, and those from power,
industry and transportation based on the spatial distribution of gross domestic product
(GDP). The $NO_X$ emissions from soil were originally obtained from Yienger and Levy
(1995) and were doubled as advised by Zhao and Wang (2009). The emissions of Cl,
HCl and lightning $NO_X$ were collected from the Global Emissions Initiative (GEIA,
Price et al., 1997). Biogenic emissions were derived from the Model Emissions of
Gases and Aerosols from Nature developed under the Monitoring Atmospheric
Composition and Climate project (MEGAN MACC, Sindelarova et al., 2014).
Meteorological fields were provided by the Weather Research and Forecasting
Model (WRF) version 3.4, a state-of-the-art atmospheric modeling system designed
for both numerical weather prediction and meteorological research (Skamarock et al.,
2008). The simulated parameters from WRF for D2 in January, April, July and
October of 2016 were compared with the observation dataset of US National Climate
Data Center (NCDC), as summarized in Table S1 in the Supplement. The index of
agreement (IOA) of wind speed for the four months between the two datasets was
larger than 0.8. The Root Mean Square Error (RMSE) of wind directions for the four
months was smaller than $40°$, and the index of agreement (IOA) of temperature and
Relative humidity between the two datasets was larger than 0.8 and 0.7, respectively.



The simulated meteorological parameters in D2 could reach the benchmarks derived
from Emery et al. (2001) and Jiménez et al. (2006).

The hourly $NO_2$ and $O_3$ concentrations were observed at 230 state-operated

stations of air quality monitoring in 41 cities within the YRD region, and they were
applied to evaluate the model performance. Locations of the stations are indicated in
Figure 1, and the observation data were derived from the China National
Environmental Monitoring Center (http://www.cnemc.cn/). The observations of $SO_4^{2-}$,
$NO_3^-$ and $NH_4^+$ (SNA) concentrations in $PM_{2.5}$ for the YRD region during 2015-2017
were collected and applied to evaluate the influence of the top-down estimation of
$NO_X$ emissions on SNA simulation. In particular, the hourly SNA concentrations of
$PM_{2.5}$ at Jiangsu Provincial Academy of Environmental Science, an urban site in the
capital city of Jiangsu Province, Nanjing (JSPAES; Chen et al., 2019b), were
observed with the Monitor for Aerosols and Gases in ambient Air (MARGA;
Metrohm, Switzerland) for January, April, July and October 2016. Meanwhile, the
daily average concentrations of SNA were also available from MARGA measurement
for the four months at the Station for Observing Regional Processes and the Earth
System, a suburban site in eastern Nanjing (SORPES; Ding et al., 2019). Besides, the
seasonal average concentrations of SNA were available at another four sites in YRD,
including the Nanjing University of Information Science & Technology site in
Nanjing (NUIST, Zhang, 2017), and three sites respectively in the cities of Hangzhou
(HZS; Li, 2018), Changzhou (CZS; Liu et al., 2018) and Suzhou (SZS; Wang et al.,
2016). Details of the collected SNA measurement studies are summarized in Table S2
in the supplement, and the locations of those sites are illustrated in Figure 1.
**2.3 Scenario setting of sensitivity analysis**

In general, there are two categories of chemical regimes (VOC-limited and

NOx-limited) in $O_3$ formation (Wang et al., 2009; Jin et al., 2017). In the VOC-limited
regime, growth in $O_3$ concentrations occurs with increased VOCs emissions and
declined $NO_X$ emissions, while the increased $NO_X$ emissions result in enhancement of
$O_3$ concentrations in the NOx-limited regime. To explore the sources and potential





control approaches of $O_3$ pollution, the sensitivity of $O_3$ formation to its precursor
emissions was analyzed with CMAQ modeling in the YRD region. As summarized in
Table S3 in the supplement, eight cases were set besides the base scenario with the
top-down $NO_X$ estimates for April 2016, the month with the largest $O_3$ concentration
observed during the research period. Cases 1 and 2 reduced only the $NO_X$ emissions
by 30% and 60%, and Cases 3 and 4 reduced only the $VOC_S$ emissions by 30% and
60%, respectively. To explore the co-effect of VOCs and $NO_X$ emission controls on
$O_3$ concentrations, Cases 5-8 with different reduction rates of VOCs and $NO_X$
emissions were designed. The emissions of $NO_X$ and VOCs in Case 5 were decreased
by 30% and 60%, and in Case 6 by 60% and 30%, respectively. Both $NO_X$ and VOCs
emissions were reduced 30% and 60% in Cases 7 and 8, respectively.

The response of SNA concentrations to the changes in precursor emissions was

influenced by various factors including the abundance of $NH_3$, atmospheric oxidation,
and the chemical regime of $O_3$ formation (Wang et al., 2013; Cheng et al., 2016; Zhao
et., 2020). To explore the sensitivity of SNA formation to its precursor emissions, four
cases were set besides the base scenario for January 2016, the month with the largest
observed SNA concentrations. As shown in Table S4 in the supplement, the emissions
of $NO_X$, $SO_2$ and $NH_3$ were reduced by 30% in Cases 9-11, respectively, and the
emissions of $NO_X$, $SO_2$ and $NH_3$ were simultaneously decreased by 30% in Case 12.

### 3.    Results and discussion

**3.1 Evaluation of the bottom-up and top-down estimates of $NO_X$ emissions**

Figure 2 compares the magnitude of the $NO_X$ emissions estimated based on the

bottom-up (MEIC) and top-down methods by month in the YRD region. The
top-down estimates were smaller than the bottom-up ones for all the concerned four
months, and the average of the monthly $NO_X$ emissions were calculated at 260.0
Gg/month for 2016 with the top-down method, 24% smaller than the bottom-up
estimation. The comparison indicates a probable overestimation in $NO_X$ emissions
with current bottom-up methodology, attributed partly to the insufficient consideration
of the effect of recent control on emission abatement. Stringent measures have





gradually been conducted to improve the local air quality in the YRD region. For
example, the "ultra-low" emission policy for power sector started in 2015, requiring
the $NO_X$ concentration in the flue gas of coal-fired unit the same as that of gas-fired
unit. The technology retrofitting on power units have been widely conducted,
significant improving the $NO_X$ removal efficiencies of selective catalytic reduction
(SCR) systems. Those detailed changes in emission control, however, could not be
fully and timely incorporated into the national emission inventory that relied more on
the routinely reported information and policy of environmental management over the
country. With the on-line data from continuous emission monitoring systems (CEMS)
incorporated, the $NO_X$ emissions from power sector were estimated to be 53% smaller
than MEIC for the China in 2015 in our previous work (Zhang et al., 2019). The bias
between the top-down and bottom-up estimates could be larger in earlier years and
reduced more recently. According to Yang et al. (2019b) and Qu et al. (2017), for
example, the top-down $NO_X$ emissions were 44% and 31% smaller than bottom-up
ones for the YRD region and the whole China in 2012. Benefiting from the better data
availability, the bottom-up inventory has been improved with the inclusion of more
information on individual power and industrial plants for recent years (Zheng et al.,

2018).

The differences in the spatial distribution of $NO_X$ emissions between the

bottom-up and top-down estimates are illustrated by month for the YRD in Figure S2
in the supplement. The top-down estimates were commonly smaller than the
bottom-up ones in the east-central YRD with intensive manufacturing industry and
population, and larger than those in most of Zhejiang Province with more hilly and
suburban regions. The bias might result from following issues. From a bottom-up
perspective, on one hand, more stringent control measures were preferentially
conducted for power and industrial plants in regions with heavier air pollution like
east-central YRD. As mentioned above, the effects of such actions were difficult to be
fully tracked in the bottom-up inventory, leading to the overestimation in emissions
for those regions. Due to the lack of precise locations of individual industrial plants
(except for large point sources), moreover, the spatial allocation of the emissions





relied commonly on the densities of population and economy, assuming a strong
correlation with emissions for them. Such assumption, however, would not still hold
in recent years, as a number of factories in the relatively developed region were
moved to the less developed suburban regions (e.g., southern Zhejiang) for both
environmental and economic purposes. The insufficient consideration of the movings
of emission sources was thus expected to result in overestimation in emissions for
developed regions and underestimation for the less developed. On the other hand, the
satellite-derived TVCDs were relatively small in southern Zhejiang (Fig. S1), and
larger error in satellite retrieval and thereby emission constraining with the inversed
modeling was expected.

Figure 3 illustrates the observed and simulated hourly $NO_2$ concentrations using

the bottom-up and top-down estimates of $NO_X$ emissions in the CMAQ by month.
The $NO_2$ concentrations simulated with the bottom-up estimates were clearly larger
than the observation in all the four concerned months, with the largest and smallest
normalized mean bias (NMB) reaching 111% and 34 % for July and January,
respectively. The result suggests again the overestimation in $NO_X$ emissions in the
current bottom-up inventory for the YRD. The model performance based on the
top-down estimates was much better than that based on the bottom-up ones, indicating
that the inversed modeling with satellite observation constraint effectively improved
the estimation of $NO_X$ emissions. The biggest improvement was found for July, with
the NMB reduced from 111% to -0.4% and the NME reduced from 111% to 33%. As
shown in Fig. 2, relatively big reduction from the bottom-up to top-down estimation
in $NO_X$ emissions was found for July compared to most of other months.

Scatter plots of the annual means of the observed and simulated surface $NO_2$

concentrations are shown in Figure S3 in the supplement. The slope between the
observation and simulation with the top-down estimate (0.99) was much closer to 1
than that with the bottom-up one (1.57), indicating clearly the advantage of the
top-down method on the constraining of the magnitude of the total emissions in the
YRD region. The difference in the two slopes implies that the surface $NO_2$
concentrations simulated with the bottom-up estimation were over 50% larger than





those based on top-down ones. As a comparison, the total emissions in the bottom-up
inventory were only 30% larger than the top-down estimation for the whole YRD
region. The larger overestimation in the concentrations than the emissions from the
bottom-up inventory could result partly from the bias of the locations of
state-operated ground observation sites. Most of those sites were located in the urban
areas where excess emissions were allocated according to the high density of
economy and population, and elevated concentrations were thus simulated compared
to rural areas. The similar correlation coefficients (R) suggested that the spatial
distribution of $NO_X$ emissions was not greatly improved in the top-down estimation
on an annual basis of urban observation. Uncertainty existed in the satellite
observation: the NMB between $NO_2$ TVCDs in POMINO and available ground-based
MAX-DOAS observations was 21% in cloud-free days (Liu et al., 2019). Due mainly
to the $NO_X$ transport, moreover, a bias of 13%-33% on the spatial distribution of
emissions was estimated for the inversed method at the horizontal resolution of 9 km
or finer (Yang et al., 2019b). Inclusion of more available observation in rural areas
helps improve the comprehensive evaluation of emission estimation.
Figure 4 illustrates the spatial distribution of monthly mean $NO_2$ concentrations
simulated based on the top-down estimates and the differences between the
simulations with the top-down and bottom-up ones. The larger $NO_2$ concentrations
existed in the east-central YRD for all the months (left column in Fig. 5), and the
difference in spatial distribution of $NO_2$ concentrations (right column in Fig. 5) was
similar with that in $NO_X$ emissions (Fig. S2). Larger reduction in $NO_2$ concentrations
based on the top-down estimates was commonly found in east-central YRD, while the
increased concentrations were found in most of Zhejiang.
**3.2 Evaluation of the $O_3$ simulation based on the top-down $NO_X$ estimates**
Figure 5 shows the observed and simulated hourly $O_3$ concentrations based on
the bottom-up and top-down estimates of $NO_X$ emissions by month. Indicated by the
smaller NMBs and NMEs, the model performance of $O_3$ based on the top-down
estimates was better than that based on the bottom-up ones for most months. It



suggests that the constrained $NO_X$ emissions with satellite observation could play an
important role on the improvement of $O_3$ simulation. The largest improvement was
found in January, for which the NMB and NME were changed from -44% and 49% to
13% and 40%, respectively, attributed to the biggest change in $NO_X$ emissions
between the top-down and bottom-up estimates for the month. The worse $O_3$
modeling performance was found for July when the top-down estimate instead of the
bottom-up one was applied in the simulation, indicated by the increased NMB and
NME. Besides the changed $NO_X$ emissions, the worse $O_3$ simulation might result as
well from the uncertainty in emissions of the volatile organic compounds (VOCs) and
the chemical mechanism of AQM in summer. As suggested by Li (2019), the biogenic
VOCs (BVOCs) emissions of the YRD region could be overestimated by 121% in
summer attributed to ignoring the effect of droughts, and such overestimation might
elevate the $O_3$ concentrations in AQM. In order to explore the influence of uncertainty
of BVOCs emissions on $O_3$ model performance, we conducted an extra case in which
the BVOCs emissions were cut by 50% in CMAQ. As shown in Figure S4 in the
supplement, the NMB between the observed and simulated $O_3$ based on the top-down
estimate of $NO_X$ emissions and the reduced BVOCs emissions declined 27% in July.
A recent study conducted an intercomparison of surface-level $O_3$ simulation from 14
state-of-the-art chemical transport models, and implied that the larger overestimation
of summer $O_3$ than winter for eastern China resulted possibly from the uncertainty in
the photochemical treatment in models (Li et al., 2019).

Table 1 summarizes the observed and simulated daily maximum 8-hour averaged

(MDA8) $O_3$ concentrations based on the bottom-up and top-down estimates of $NO_X$
emissions are summarized by month for the YRD region. The MDA8 $O_3$
concentrations simulated with the top-down estimates were larger than those with the
bottom-up ones, and were closer to the observation for most months. As most of the
YRD was identified as the VOC-limited region (Li et al., 2012; Zhou et al., 2017), the
reduced $NO_X$ emissions with the top-down method enhanced the $O_3$ levels in the
AQM. Similar to the hourly concentrations, the most significant improvement for
MDA8 was found in January, with the NMB and NME reduced from -35% and 39%



to 11% and 28%, respectively. Moreover, the improvement of April and October for
MDA8 was larger than that for the hourly concentrations, indicating that the improved
$NO_X$ emissions were more beneficial for the simulation of daytime peak $O_3$
concentrations in spring and winter. Figure 6 illustrates the spatial distribution of the
monthly mean $O_3$ concentrations simulated based on the top-down $NO_X$ estimates and
the differences between the simulations with the top-down and bottom-up estimates
by month. In contrast to $NO_2$, the smaller $O_3$ concentrations existed in the east-central
YRD for most months, as it was identified as the VOC-limited region with relatively
high $NO_2$ level. Larger $O_3$ concentrations were found for the surrounding regions in
the YRD, e.g., southern Zhejiang, attributed partly to the relatively abundant BVOC
emissions. An exception existed for July, with clearly larger $O_3$ concentrations in
east-central YRD. With the largest population density and most developed economy in
YRD, the area contains a large number of chemical industrial plants and solvent
storage, transportation and usage (Zhao et al., 2017). High temperature in summer
promoted the volatilization of chemical products and solvent, and thereby enhanced
the seasonal VOCs emissions more significantly compared to other less developed
YRD regions. Moreover, the lowest $NO_2$ concentration found in summer helped
increase the $O_3$ concentration for the region (Gu et al., 2020). Regarding the
simulation difference with two emission estimates, application of the top-down
estimates instead of the bottom-up ones elevated the $O_3$ concentrations in most of the
YRD region. In particular, the big reduction in $NO_X$ emissions for the east-central
YRD (Figure S2) resulted in the more evident growth in $O_3$ concentrations, reflecting
the negative effect of $NO_X$ abatement on $O_3$ pollution control in the VOC-limited
regions.
**3.3 Evaluation of SIA simulation based on the top-down $NO_X$ estimates**

Shown in Table 2 is the comparison between the observed and simulated SNA

($SO_4^{2-}$, $NO_3^-$ and $NH_4^+$) concentrations by season. Larger observed and simulated
SNA concentrations were found in winter and spring, and smaller were found in
summer and autumn. For most seasons, the simulations of $NO_3^-$ concentrations were





moderately improved with the top-down estimates of $NO_X$ emissions for all the
concerned YRD cities, with an exception of Nanjing in autumn. The largest
improvement was found in summer, with the mean bias between the simulation and
observation reduced 35% for all the involved cites. Compared to the bottom-up
inventory, the commonly smaller $NO_X$ emissions in the top-down estimates limited
the $NO_2$ concentration and suppressed the formation of $NO_3^-$, while the enhanced $O_3$
from the reduced $NO_X$ emissions promoted it (Cai et al., 2017; Huang et al., 2020). In
summer, the former dominated the process with the most evident improvement in $NO_2$
simulation (Figure 3), thus the reduced $NO_3^-$ concentrations that were closer to
observation were simulated for all the cities.

The simulations with both top-down and bottom-up estimates of $NO_X$ emissions

underestimated the $NH_4^+$ concentrations for most cases, and such underestimation was
slightly corrected with the application of the top-down estimates except for summer.
The average change in $NH_4^+$ concentrations was 2.3%, much smaller than that of
$NO_3^-$ at 14%. The moderate improvement in $NH_4^+$ simulation with the reduced $NO_X$
emissions in the top-down estimates resulted partly from the enhancement of the
simulated $O_3$ concentrations and thereby the promoted $NH_4^+$ formation. In summer,
however, the significant drop in the simulated $NO_2$ concentration was assumed to
reduce the $NO_3^-$ and $NH_4^+$ formation, and to weaken the consistency between the
simulated and observed $NH_4^+$. The difference between the simulated $SO_4^{2-}$ with the
bottom-up and top-down $NO_X$ emission estimates were small for most seasons,
implying a limited benefit of improved $NO_X$ emissions on $SO_4^{2-}$ modeling.

Figure 7 shows the differences in the spatial distribution of SNA concentrations

simulated with the bottom-up and top-down estimates of $NO_X$ emissions by month. In
most of the region, the differences of $NO_3^-$ concentrations were larger than those of
$NH_4^+$ and $SO_4^{2-}$ for all seasons, and they were mainly controlled by the changed
ambient $NO_2$ or $O_3$ level. The difference in spatial pattern of $NO_3^-$ was similar to that
of $O_3$ for January, and the larger growth attributed to the application of the top-down
estimates was found in northern Anhui and eastern Zhejiang (Fig. 7a). The result
implies that the change in $NO_3^-$ concentration in winter could result partly from the





improved $O_3$ simulation, i.e., the elevated $O_3$ was an important reason for the
enhanced the formation of SNA in winter (Huang et al., 2020). Similarly, the
increased $NO_3^-$ was found for more than half of the YRD region in April, along with
the growth of $O_3$ concentrations (Fig. 7d). For July, however, the difference in spatial
pattern of $NO_3^-$ (Fig. 7g) was similar with $NO_2$ (Fig. 4g), and the larger reduction
attributed to the application of the top-down estimates was found in northern YRD.
The result suggests that the declining $NO_X$ emissions and thereby $NO_2$ concentration
dominated the reduced $NO_3^-$ formation in summer. In October, the growth in $NO_3^-$
concentrations was found again in most YRD when the top-down estimates were
applied (Fig. 7j). The growth in the north resulted mainly from the increased $O_3$ level,
while that in the south was associated with the increased $NO_2$. The differences in
spatial patterns of simulated $NH_4^+$ concentrations were similar to those of $NO_3^-$ for the
four months, suggesting that the change in $NH_4^+$ was associated with formation and
decomposition of $NH_4NO_3$. However, the changes of spatial distribution of $SO_4^{2-}$ were
similar with those of $O_3$ concentration. Since $NH_4^+$ was preferred to react with $SO_4^{2-}$
rather than $NO_3^-$ (Wang et al., 2013), the formation of $SO_4^{2-}$ was mainly influenced by
the atmospheric oxidizing capacity when only $NO_X$ emissions were changed.
Figure 8 illustrates the observed and simulated hourly $NO_3^-$ concentrations based
on the bottom-up and top-down estimate of $NO_X$ emissions by month at JSPAES. The
NMBs and NMEs for simulation with the top-down emissions were smaller than those
with bottom-up ones in January and July, implying the benefit of the improved $NO_X$
emissions on hourly $NO_3^-$ concentration simulation in winter and summer. The best
model performance with the top-down estimates was found in January, with the
hourly variation commonly caught with AQM. However, the $NO_3^-$ concentration was
seriously overestimated and the model failed to catch the hourly variations in summer
indicated by the large NMB and NME. As shown in Figure S5 in the supplement, both
the $NO_2$ and $O_3$ concentrations at JSPAES were significantly overestimated for July
except $O_3$ with the bottom-up $NO_X$ emission estimate, and it partly explained the
elevated $NO_3^-$ level from CMAQ simulation.
Figures S6 and S7 in the supplement compare the observed and simulated hourly





concentrations at JSPAES by month for $NH_4^+$ and $SO_4^{2-}$, respectively. The NMBs and
NMEs for $NH_4^+$ simulation with the top-down estimates were smaller than those with
the bottom-up ones for most months, while the changes in $SO_4^{2-}$ concentration were
small. The $NH_4^+$ and $SO_4^{2-}$ concentrations were largely underestimated with the
top-down estimates in January, indicated by the NMB at -44% and -38%, respectively.
Meanwhile, as shown in Figure S8 in the supplement, the $SO_2$ concentrations were
overestimated by 61% at the site. The results thus imply a great uncertainty in the
gas-particle partitioning of $(NH_4)_2SO_4$ formation in the model in winter, attributed
probably to the missed oxidation mechanisms of $SO_2$ (Chen et al., 2019c).

**3.4 Sensitivity analysis of $O_3$ and SNA formation in the YRD region**

Table 3 summarizes the relative changes in the simulated $O_3$ concentrations for
April 2016 in different cases. The mean $O_3$ concentration would decline by 8.9% and
19.5% with 30% and 60% VOCs emissions off (Cases 2 and 7), while it would
increase by 14.2% and 23.7% with 30% and 60% $NO_X$ emissions off (Cases 1 and 6),
respectively. The result confirmed the VOC-limited regime of $O_3$ formation in the
YRD region: controlling VOCs emissions was an effective way to alleviate $O_3$
pollution, while reducing $NO_X$ emissions alone would aggravate $O_3$ pollution.
The growth of $O_3$ concentrations was also found when the reduction rate of $NO_X$
emissions was equal to or larger than that of VOCs. The $O_3$ concentration would
increase by 7.1% and 14.5% respectively when both $NO_X$ and VOCs emissions were
reduced by 30% and 60% (Cases 3 and 8), and it would increase by 19.8% when $NO_X$
and VOCs emissions were respectively declined by 60% and 30% (Case 5). In
contrast, small abatement of $O_3$ concentrations (2.1%) was achieved from the 30%
and 60% reduction of emissions respectively for $NO_X$ and VOCs (Case 4), implying
that the $O_3$ level could be restrained when the reduction rate of VOCs was twice of or
more than that of $NO_X$. To control the $O_3$ pollution effectively and efficiently,
therefore, the magnitude of VOCs and $NO_X$ emission reduction should be carefully
planned and implemented. In actual fact, controlling VOCs is more difficulty than
$NO_X$. Compared to $NO_X$ that comes mainly from fossil fuel combustion (Zheng et al.,





2018), it is more complicated to identify the sources of specific VOCs species that are
most active in $O_3$ formation (Wei et al., 2014; Zhao et al., 2017). Moreover,
substantial VOC emissions are from area or fugitive sources, for which the emission
control technology can hardly be effectively applied. Therefore, it is a big challenge to
control $O_3$ pollution by reducing more VOCs than $NO_X$.

Figure 9 illustrates the differences in spatial patterns of the simulated monthly

mean $O_3$ concentrations between the base and sensitivity cases in April. The $O_3$
concentrations were expected to decline for the whole YRD region in the cases of
30% and 60% VOCs emissions off (Fig. 9b and 9d), indicating the VOC-limited
regime of $O_3$ formation for the entire YRD. For other cases, the $O_3$ concentrations
were clearly elevated in the central-eastern YRD with relatively large population and
developed industry, particularly for the cases with $NO_X$ control only (Fig. 9a and 9c)
or relatively large $NO_X$ abatement together with VOC control (Fig. 9f and 9g). Even
for the case with 60% of VOCs reduction and 30% of $NO_X$ (Fig. 9h), there was still
small increase in $O_3$ concentration in central-eastern YRD, in contrast to the slight $O_3$
reduction found for most of YRD areas. Those results reveal the extreme difficulty in
$O_3$ pollution control for the region. In southwestern Zhejiang, the $O_3$ concentrations
were found to decline in the cases with large abatement of $NO_X$ emissions (Fig. 9b, 9f
and 9g), suggesting a shifting from VOC-limited to $NO_X$ limited region for the $O_3$
formation.

Table 4 summarizes the change in the simulated monthly means of SNA ($NO_3^-$,

$NH_4^+$ and $SO_4^{2-}$) concentrations between the base case and sensitivity cases in January.
The SNA concentrations were decreased in most cases, implying that the reduction in
precursor emissions was useful for mitigating the SNA pollution. Compared to that of
precursor emissions, however, the reduction rate of SNA was much smaller attributed
to the strong nonlinearity of SNA formation. The largest reductions were found at
11.7% and 12.4% when emissions of $NH_3$ and all the three precursors were decreased
by 30% (Cases 11 and 12), respectively. In contrast, the SNA concentrations declined
slightly by 1% and increased by 0.5% when $NO_X$ and $SO_2$ emissions were reduced by
30% (Cases 9 and 10), respectively. The results suggest that most of YRD was in an





NH$_3$-neutral or even NH$_3$-poor condition in winter, consistent with the judgment
through AQM based on an updated NH$_3$ emission inventory (Zhao et al., 2020), as the
NH$_3$ volatilization in winter was much smaller than other seasons. Reducing NH$_3$
emissions was the most efficient way to control SNA pollution for the region in winter.
In Case 11 with NH$_3$ control only, the reduced NO$_3^-$ and NH$_4^+$ were much larger than
that of SO$_4^{2-}$. As NH$_3$ reacted with SO$_2$ prior to NO$_X$, NH$_4$NO$_3$ was assumed easier to
decompose than (NH$_4$)$_2$SO$_4$ when NH$_3$ emissions were reduced. The growth of NO$_3^-$
concentrations was found for Case 10 (SO$_2$ control only), since the free NH$_3$ from the
reduced SO$_2$ emissions could react with NO$_X$ in the NH$_3$-poor condition. Similarly, the
SO$_4^{2-}$ concentrations increased for Case 9 (NO$_X$ control only), as the elevated O$_3$
attributed to the reduction of NO$_X$ emissions promoted the SO$_4^{2-}$ formation.
## 4.   Summary
From a "top-down" perspective, we have estimated the monthly NO$_X$ emissions
for the YRD region in 2016, based on the nonlinear inversed modeling and NO$_2$
TVCDs from POMINO, and the bottom-up and top-down estimates of NO$_X$ emissions
were evaluated with AQM and ground NO$_2$ observation. Due to insufficient
consideration of improved controls on power and industrial sources, the NO$_X$
emissions were probably overestimated in current bottom-up inventory (MEIC),
resulting in significantly higher simulated NO$_2$ concentrations than the observation.
The simulated NO$_2$ concentrations with the top-down estimates were closer to the
observation for all the four seasons, suggesting the improved emission estimation with
satellite constraint. Improved O$_3$ and SNA simulations with the top-down NO$_X$
estimates for most months indicate the importance role of precursor emission
estimation on secondary pollution modeling for the region. Through the sensitivity
analysis of O$_3$ formation, the mean O$_3$ concentrations were found to decrease for most
YRD when only VOCs emissions were reduced or the reduced rate of VOCs was
twice of NO$_X$, and the result indicates the effectiveness of controlling VOCs
emissions on O$_3$ pollution abatement for the region. For part of southern Zhejiang,
however, the O$_3$ concentrations were simulated to decline with the reduced NO$_X$





emissions, implying the shifting from VOC-limited to $NO_X$-limited region. Compared
to reducing $NO_X$ or $SO_2$ only, larger reduction in SNA concentrations was found when
30% of emissions were cut for $NH_3$ or all the three precursors ($NO_2$, $NH_3$ and $SO_2$).
The result suggests that reducing $NH_3$ emissions was crucial to alleviate SNA
pollution of YRD in winter.

Limitations remain in this study. Due to the limited horizontal resolution of OMI,

relatively big bias existed in the spatial distribution of the constrained $NO_X$ emissions
at the regional scale compared to national or continental one, and the uncertainty
could exceed 30% for the YRD region (Yang et al., 2019b). Therefore the
improvement on the top-down estimates of $NO_X$ emissions can be expected when the
more advanced and reliable products of satellite observation get available at finer
horizontal resolution (e.g., TROPOspheric Monitoring Instrument, TROPOMI).
Besides, more SNA observations from on-line measurement are recommended for a
better space coverage and temporal resolution, to explore more carefully the response
of SNA to the changes in emissions of $NO_X$ and other precursors.

**Data availability**

All data in this study are available from the authors upon request.


**Author contributions**

YY developed the strategy and methodology of the work and wrote the draft. YZ

improved the methodology and revised the manuscript. LZ provided useful comments
on the methodology. JZ and XH provided observation data of secondary inorganic
aerosols. XZ, YZ, MX and YL provided comments on air quality modeling.

**Competing interests**

The authors declare that they have no conflict of interest.






**Acknowledgements**


This work was sponsored by Natural Science Foundation of China (91644220 and
41575142) and the National Key Research and Development Program of China
(2017YFC0210106). We would also like to thank Tsinghua University for the free use
of national emissions data (MEIC), and Peking University for the support of satellite
data (POMINO v1).

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





## FIGURE CAPTIONS

**Figure 1. The modeling domain and locations of meteorological and air quality monitoring sites. The map data provided by Resource and Environment Data Cloud Platform are freely available for academic use (http://www.resdc.cn/data.aspx?DATAID=201), © Institute of Geographic Sciences & Natural Resources Research, Chinese Academy of Sciences.**

**Figure 2. The bottom-up and top-down estimates of $NO_X$ emissions by month for the YRD region in 2016.**

**Figure 3. The observed and simulated hourly $NO_2$ concentrations based on the bottom-up and top-down $NO_X$ emissions for January, April, July and October 2016.**

**Figure 4. The spatial distribution of the simulated monthly mean $NO_2$ concentration with the top-down estimates and differences between the simulations with the top-down and bottom-up $NO_X$ emissions in January, April, July and October 2016 (top-down minus bottom-up).**

**Figure 5. The observed and simulated hourly $O_3$ concentrations with the bottom-up and top-down $NO_X$ emission estimates for January, April, July and October 2016.**

**Figure 6. The spatial distribution of the simulated monthly mean $O_3$ concentration with the top-down $NO_X$ estimates and the spatial differences between the simulations with the top-down and bottom-up $NO_X$ emissions in January, April, July and October 2016 (top-down minus bottom-up).**

**Figure 7. The spatial differences between the simulated SNA concentrations with the bottom-up and top-down $NO_X$ emission estimates for January, April, July and October 2016 (top-down minus bottom-up).**

**Figure 8. The observed and simulated hourly $NO_3^-$ concentrations with the bottom-up and top-down $NO_X$ emission estimates for January, April, July and**





**October 2016 at JSPEAS.**

**Figure 9. The spatial differences of monthly mean O₃ concentrations between the simulations based on base case (top-down estimates) and sensitivity cases in April 2016 (sensitivity case minus base case).**





**Table 1. The model performance statistics of daily maximum 8-hour averaged (MDA8) $O_3$ concentrations in January, April, July and October 2016 with the bottom-up and top-down $NO_X$ emissions.**

| Month | Emission input | Observed ($\mu g/m^3$) | Simulated ($\mu g/m^3$) | NMB | NNE |
|---|---|---|---|---|---|
| January | Bottom-up | 50.6 | 33.0 | -34.8% | 38.6% |
| | Top-down | | 56.3 | 11.3% | 27.7% |
| April | Bottom-up | 101.5 | 87.2 | -14.1% | 20.2% |
| | Top-down | | 108.5 | 6.9% | 16.1% |
| July | Bottom-up | 107.4 | 117.3 | 9.2% | 15.7% |
| | Top-down | | 140.7 | 31.0% | 31.0% |
| October | Bottom-up | 65.9 | 53.9 | -18.3% | 23.2% |
| | Top-down | | 73.4 | 11.3% | 21.7% |



**Table 2. Comparison of observed and simulated $NO_3^-$, $NH_4^+$ and $SO_4^{2-}$ concentrations by site and season in 2016 (unit: μg/m$^3$). The information of SNA observation sites is provided in Table S2 in the supplement. BU and TD indicate the CMAQ modeling with the bottom-up and top-down estimate of $NO_X$ emissions, respectively.**

| | Spring | | | Summer | | | Autumn | | | Winter | | |
|---|---|---|---|---|---|---|---|---|---|---|---|---|
| | $NO_3^-$ | $NH_4^+$ | $SO_4^{2-}$ | $NO_3^-$ | $NH_4^+$ | $SO_4^{2-}$ | $NO_3^-$ | $NH_4^+$ | $SO_4^{2-}$ | $NO_3^-$ | $NH_4^+$ | $SO_4^{2-}$ |
| JSPAES | 19.1 | 16.5 | 12.7 | 5.7 | 9.3 | 10.5 | 10.3 | 6.1 | 9.7 | 31.1 | 16.5 | 20.3 |
| CMAQ (BU) | 20.7 | 8.5 | 12.0 | 14.4 | 6.0 | 9.1 | 10.9 | 5.0 | 9.0 | 25.6 | 9.3 | 12.8 |
| CMAQ (TD) | 22.3 | 9.0 | 12.2 | 11.8 | 5.4 | 9.5 | 11.6 | 5.2 | 9.1 | 26.2 | 9.4 | 12.8 |
| SORPES | 14.1 | 8.6 | 13.2 | 7.5 | 6.6 | 11.5 | 8.8 | 5.2 | 8.3 | 23.0 | 13.4 | 15.7 |
| CMAQ (BU) | 18.5 | 7.3 | 8.0 | 12.2 | 4.3 | 5.2 | 9.3 | 4.0 | 5.4 | 23.6 | 8.7 | 10.9 |
| CMAQ (TD) | 18.0 | 7.0 | 7.4 | 8.3 | 3.7 | 5.0 | 9.8 | 4.2 | 5.4 | 23.6 | 8.8 | 10.1 |
| NUIST | 16.9 | 11.0 | 15.9 | 6.8 | 7.1 | 13.1 | N/A | N/A | N/A | 20.9 | 14.3 | 16.8 |
| CMAQ (BU) | 20.0 | 7.9 | 9.9 | 14.0 | 5.8 | 7.5 | | | | 24.3 | 9.0 | 11.3 |
| CMAQ (TD) | 21.8 | 8.5 | 9.9 | 11.8 | 5.3 | 7.8 | | | | 24.6 | 9.1 | 11.3 |
| HZS | 19.9 | 6.6 | 19.9 | 1.9 | 2.8 | 6.2 | 12.7 | 8.3 | 13.3 | 25.3 | 6.6 | 19.5 |
| CMAQ (BU) | 14.1 | 5.7 | 8.8 | 5.0 | 1.5 | 2.1 | 8.3 | 3.6 | 6.5 | 18.5 | 6.6 | 9.1 |
| CMAQ (TD) | 16.0 | 6.3 | 8.6 | 3.7 | 1.3 | 2.8 | 9.3 | 3.9 | 6.6 | 19.9 | 6.8 | 8.9 |
| CZS | N/A | N/A | N/A | 5.1 | 5.1 | 10.9 | N/A | N/A | N/A | 20.4 | 11.8 | 10.9 |
| CMAQ (BU) | | | | 11.6 | 4.9 | 7.1 | | | | 23.1 | 9.1 | 11.3 |
| CMAQ (TD) | | | | 10.7 | 5.0 | 7.3 | | | | 23.1 | 9.1 | 11.3 |
| SZS | 17.8 | 10.2 | 14.7 | 7.9 | 8.0 | 14.9 | 14.2 | 9.0 | 13.1 | 23.2 | 12.5 | 15.1 |
| CMAQ (BU) | 14.5 | 6.0 | 7.1 | 13.3 | 5.3 | 7.1 | 6.2 | 2.9 | 6.3 | 19.6 | 7.8 | 11.7 |
| CMAQ (TD) | 15.5 | 6.3 | 7.1 | 11.7 | 5.0 | 7.7 | 6.9 | 3.0 | 6.3 | 19.9 | 7.9 | 11.7 |
| Mean | 17.6 | 10.6 | 15.3 | 5.8 | 6.5 | 11.2 | 11.5 | 7.1 | 11.1 | 24.0 | 12.5 | 16.4 |
| CMAQ (BU) | 17.6 | 7.1 | 9.1 | 11.7 | 4.6 | 6.3 | 8.7 | 3.9 | 6.8 | 22.5 | 8.4 | 11.2 |
| CMAQ (TD) | 18.7 | 7.4 | 9.1 | 9.7 | 4.3 | 6.7 | 9.4 | 4.1 | 6.8 | 22.9 | 8.5 | 11.0 |



**Table 3. The changed percentages of ozone concentration based on the sensitivity analysis for April 2016.**

|  | No reduction | -30% VOCs emissions | -60% VOCs emissions |
|---|---|---|---|
| No reduction | - | -8.9% (Case 2) | -19.5% (Case 7) |
| -30% $NO_X$ emissions | 14.2% (Case 1) | 7.1% (Case 3) | -2.1% (Case 4) |
| -60% $NO_X$ emissions | 23.7% (Case 6) | 19.8% (Case 5) | 14.5% (Case 8) |





**Table 4. The changed percentages of $NO_3^-$, $NH_4^+$ and $SO_4^{2-}$ concentrations based on the sensitivity analysis for January 2016.**

|  | $NO_3^-$ | $NH_4^+$ | $SO_4^{2-}$ | SNA |
|---|---|---|---|---|
| -30% $NO_X$ emissions (Case 9) | -3.3% | -1.2% | 3.8% | -1.0% |
| -30% $NH_3$ emissions (Case 10) | -16.3% | -14.5% | -0.6% | -11.7% |
| -30% $SO_2$ emissions (Case 11) | 2.0% | 0.2% | -2.4% | 0.5% |
| -30% ($NO_X$+$NH_3$+$SO_2$) emissions (Case 12) | -15.5% | -15.5% | -4.0% | -12.4% |



**Figure 1.**

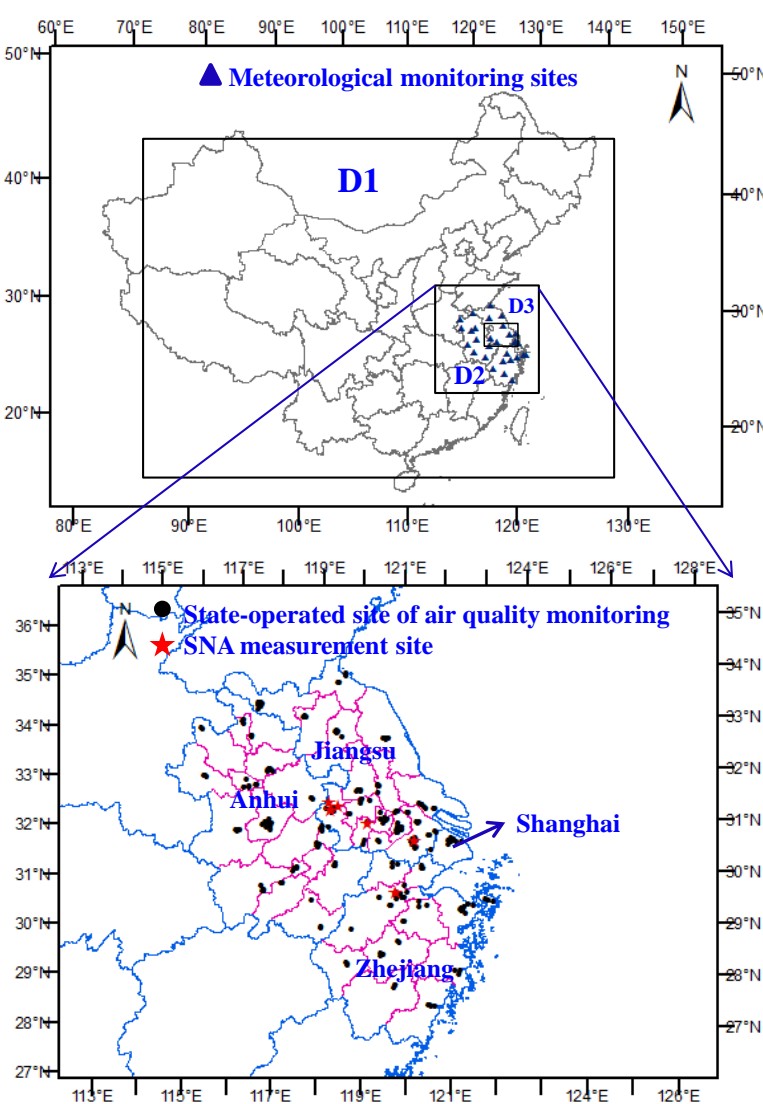





**Figure 2.**

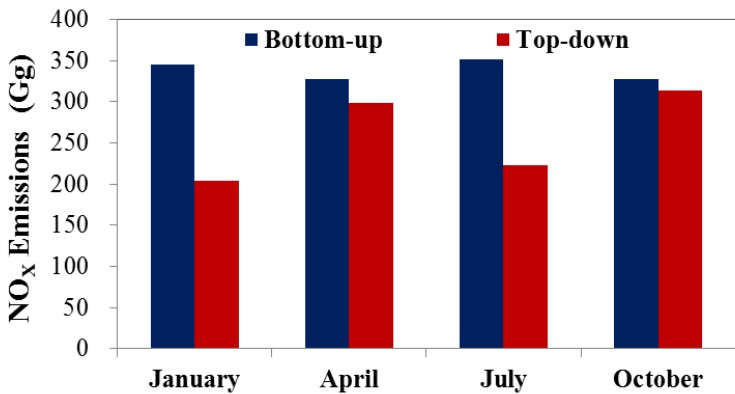





**Figure 3.**

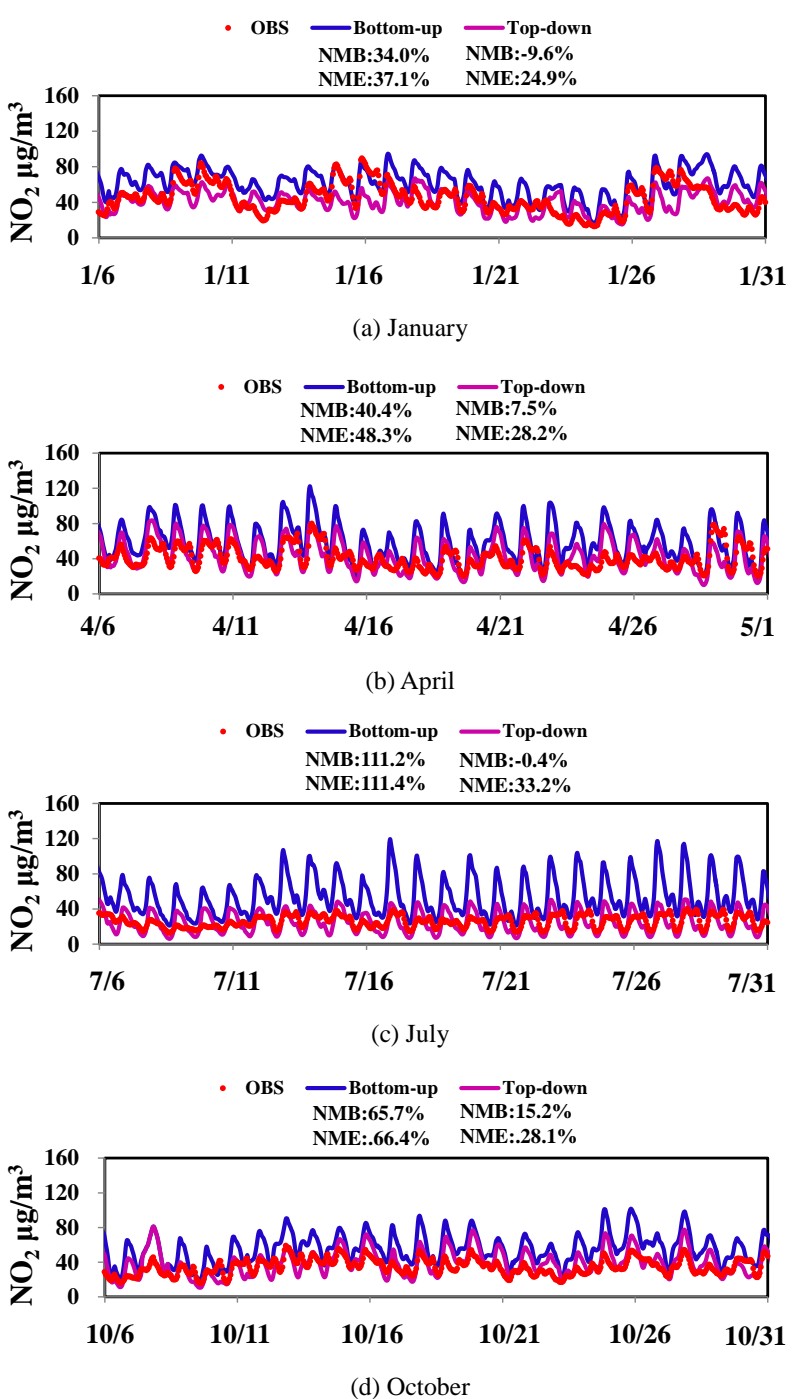

(a) January

(b) April

(c) July

(d) October





**Figure 4.**

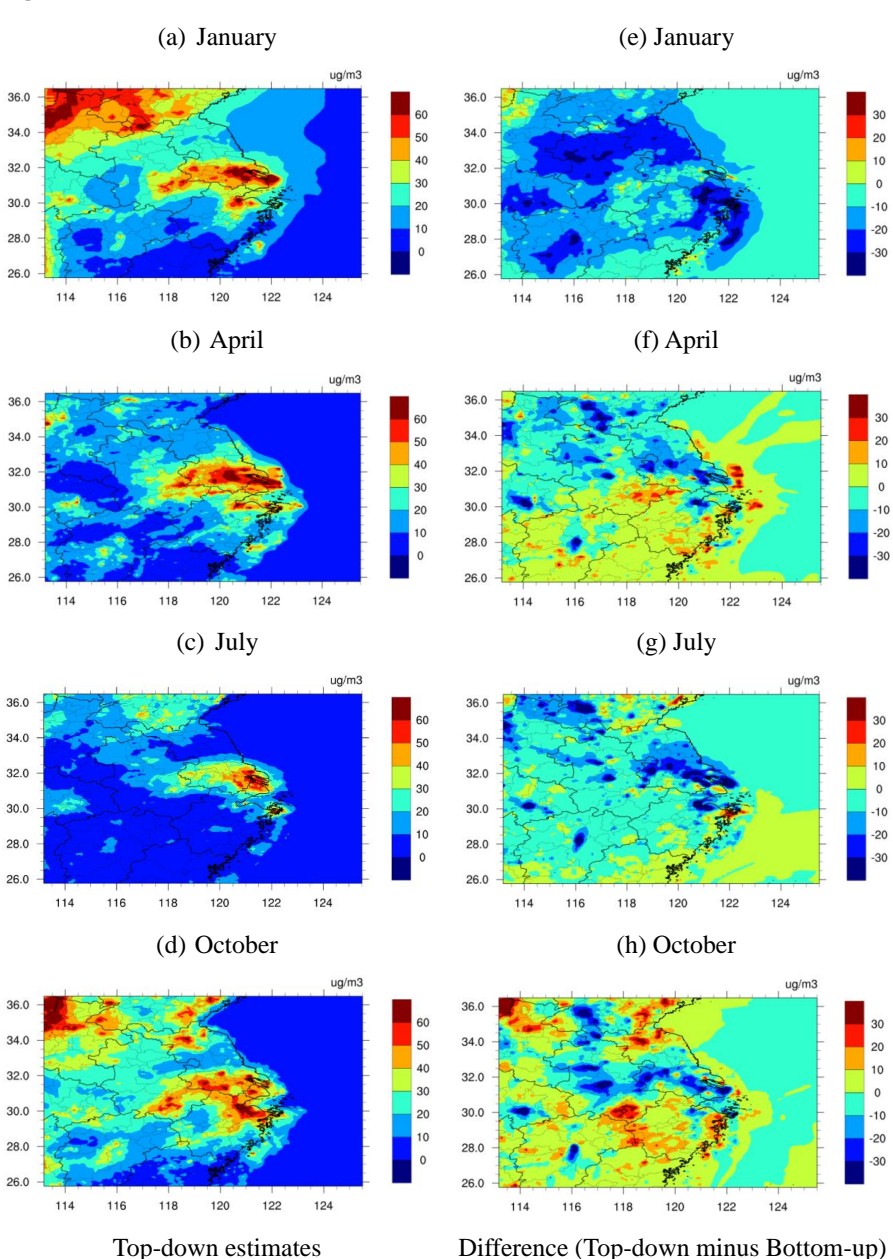

Top-down estimates          Difference (Top-down minus Bottom-up)





**Figure 5.**

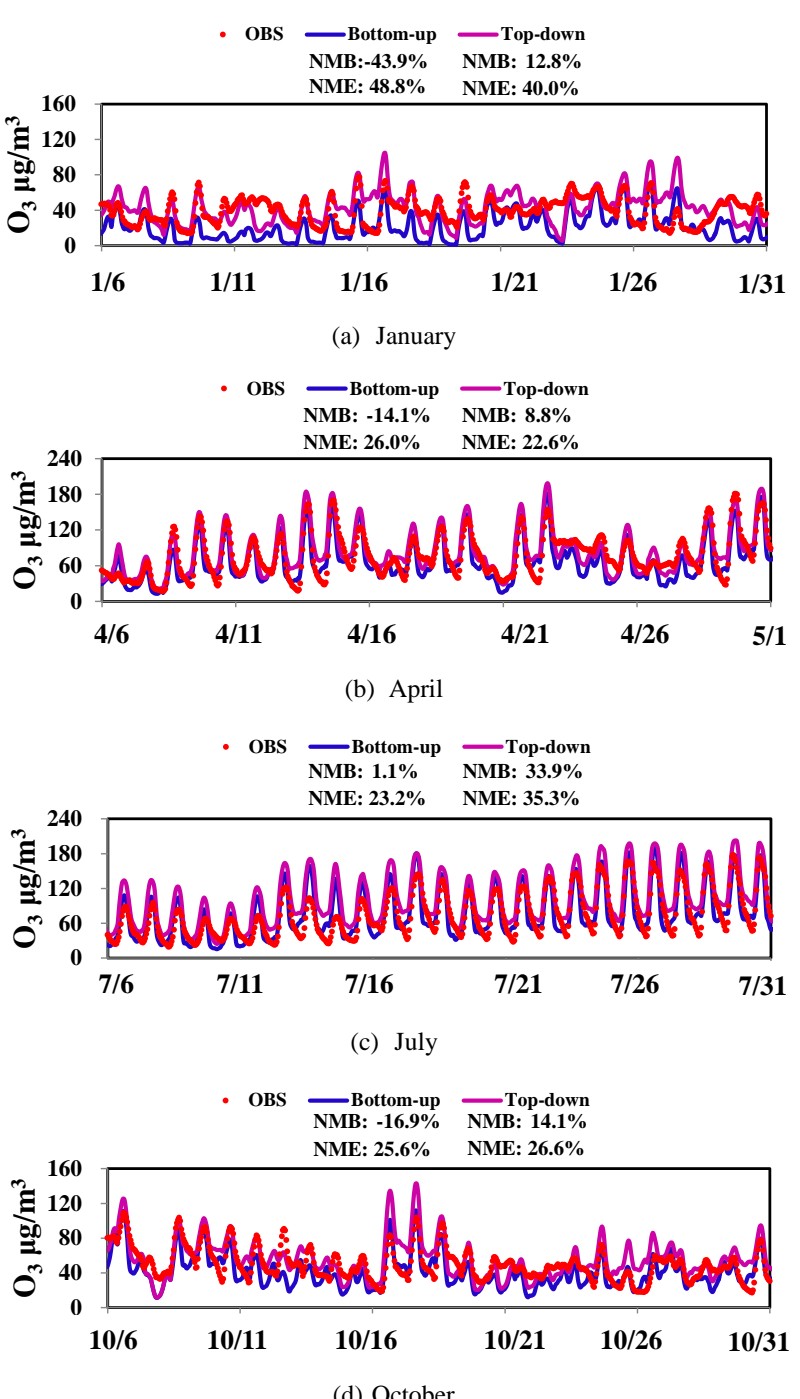

(a) January

(b) April

(c) July

(d) October




**Figure 6.**

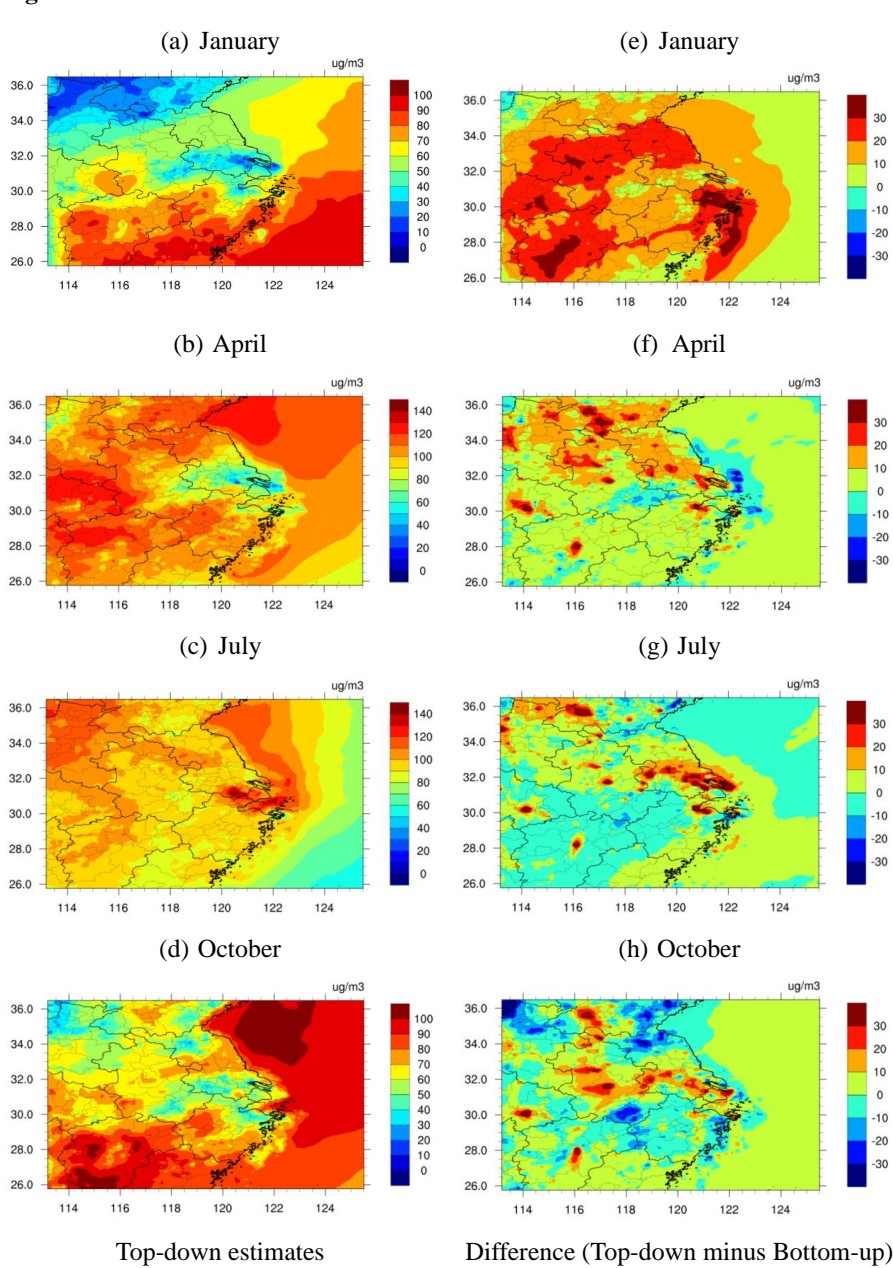





**Figure 7.**

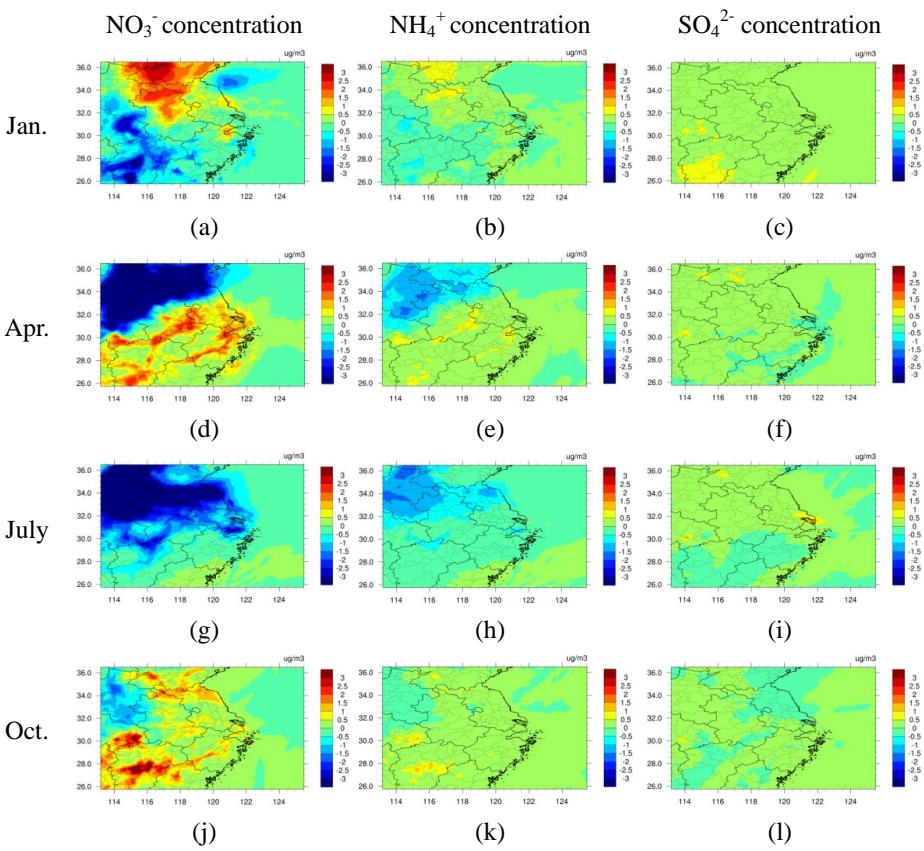





**Figure 8.**

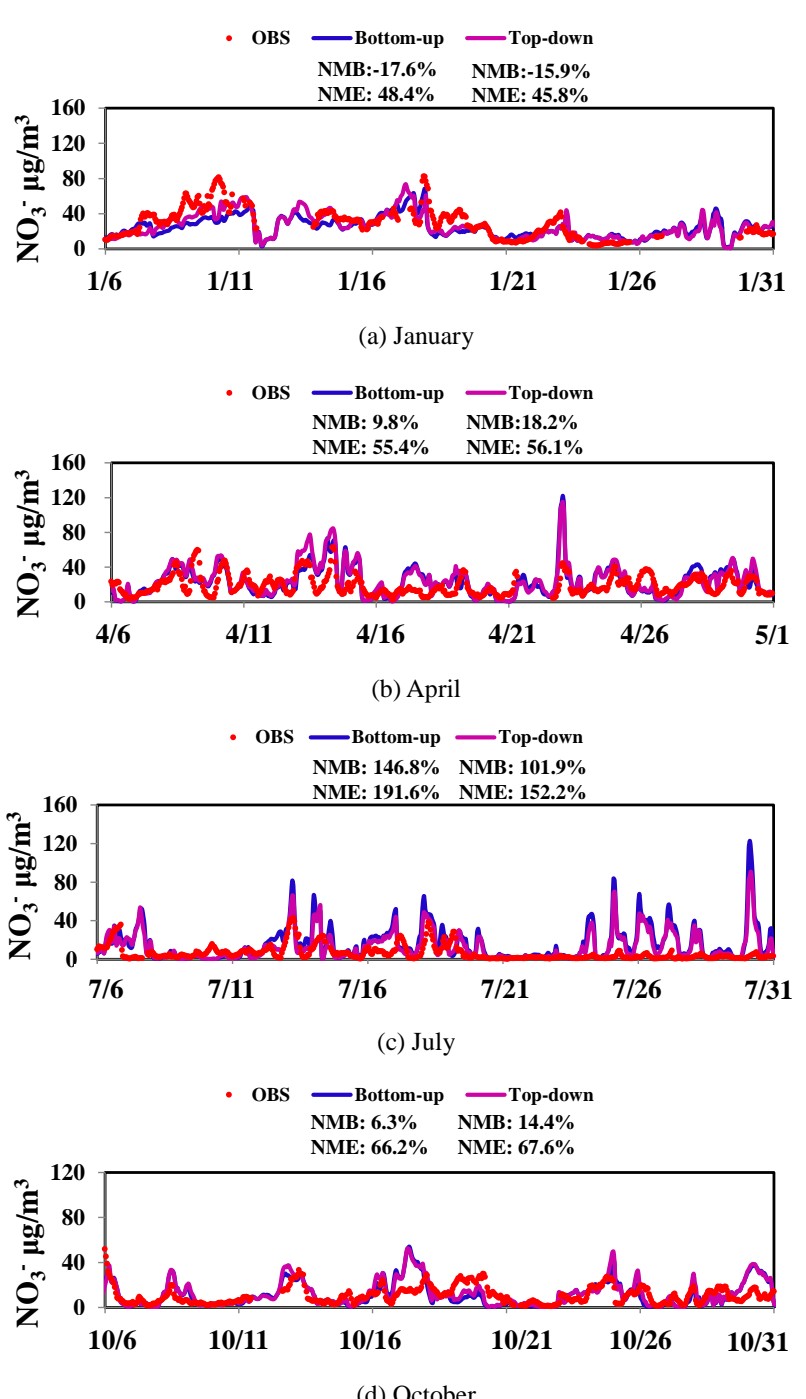





**Figure 9.**

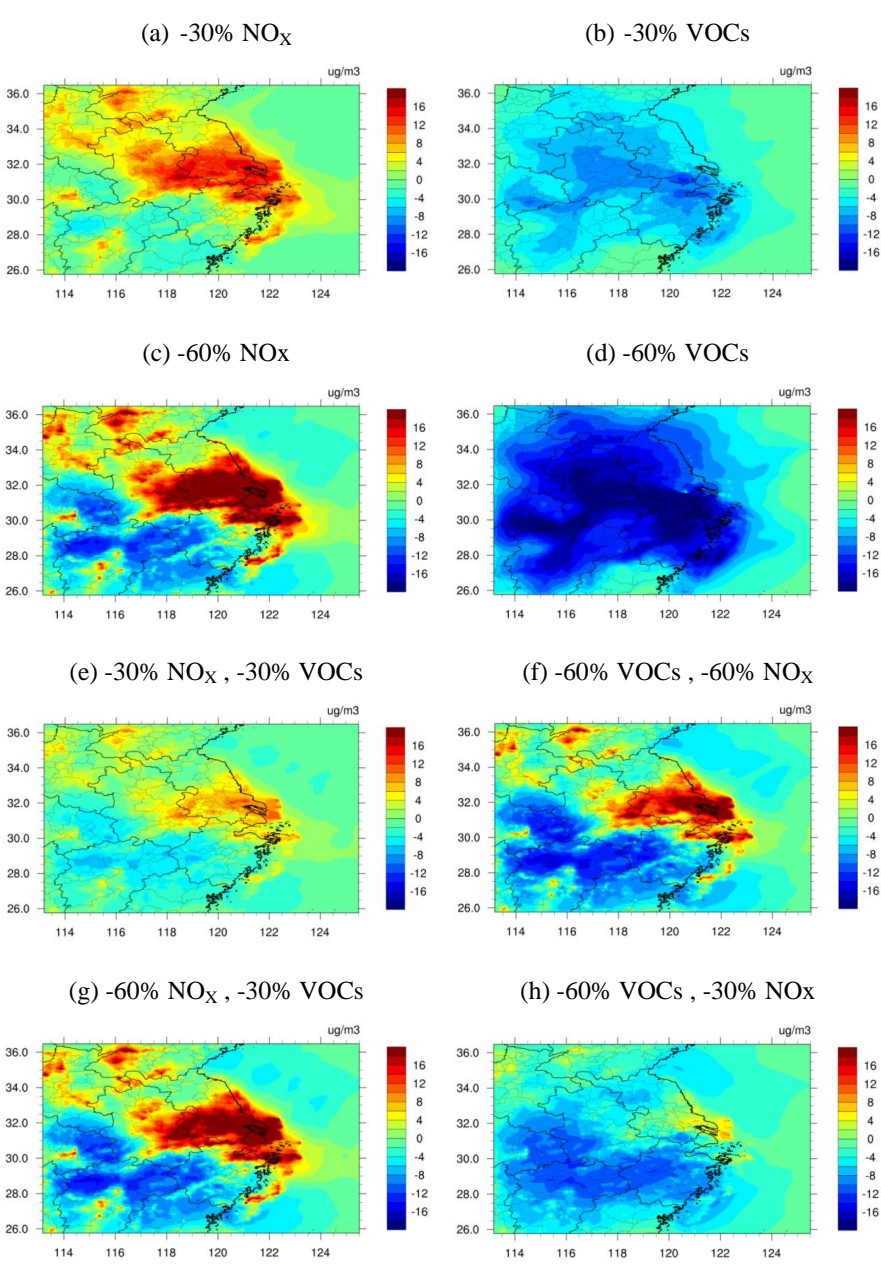