# Peer review of "Improvement from the satellite-derived $NO_X$ emissions on"

_Atmospheric Chemistry and Physics, 2020_

## Referee Comment (RC1) · Anonymous Referee #1 · 16 Sep 2020

The authors developed a "top-down" methodology based on the inversed chemistry-transport modeling and satellite data to estimate the NOx emissions for four seasons in YRD region in 2016. The results show that the improved NO2, O3, and SNA simulation results can be achieved with top-down estimates comparing to current bottom-up estimates. Further sensitivity study of O3 formation indicates the effectiveness of controlling VOCs emissions on O3 pollution abatement for PRD region and reducing NH3 emissions was crucial to alleviate SNA pollution of YRD in winter. The manuscript was generally well written, the research presented is innovative and the results can

guide the policymaking. I recommend this paper to be published in ACP after some comments have been addressed. My general comments: 1. Please revised the introduction part thoroughly to improve the narrative logic, the current version is a little hard to follow and some statements need to be summarized. 2. Line 259-265: The description of Table S3 does not agree with Table S3 shown in the Supplement file. And please clarify the meaning of "-" in Table S3, preferably with a footnote. 3. Line 386-389: Why did the authors only perform an extra simulation of exploring the influence of BVOCs emissions with top-down estimate instead of with both top-down and bottom-up estimates to prove that a better O3 simulation can be achieved based on top-down NOx estimates? Please clarify it. 4. Line 409-413: Please add references after these two statements. 5. Line 423-426: Please explain more to support the inference and can authors replot figure S2? The current one is blurring. 6. Line 427: I think changing SIA to SNA would be better to keep the consistency of the full text. 7. Line 451-453: Sha et al. (2019) reported that SO2 heterogeneous oxidation can largely improve the sulfate simulation results in Nanjing. Authors may incorporate the related mechanisms to perform the simulation, if possible, or at least mention this potential reason when discussing the factors influencing the accuracy of SNA simulation. References: Sha T, Ma X, Jia H, Tian R, Chang Y, Cao F, Zhang Y. Aerosol chemical component: Simulations with WRF-Chem and comparison with observations in Nanjing. Atmospheric Environment. 2019 Dec 1;218:116982.

---

## Referee Comment (RC2) · Anonymous Referee #2 · 13 Oct 2020

This manuscript has presented a top-down estimate of NOx emissions in the Yangtze River Delta (YRD) region and demonstrated that air quality modeling using the top-down NOx emissions could improve the simulations of ozone and secondary inorganic aerosol (SIA) over this region. A set of sensitivity simulations are conducted to better understand the formation of ozone and SIA under perturbed precursor emission conditions.

This manuscript offers some new knowledge on the regional secondary pollution over YRD including an improved estimate of NOx emissions and predicted effectiveness of

various emission controls on secondary pollution formation.

This study is overall well conducted and analyzed. The manuscript is well written, and fits the scope of ACP. I think the following comments shall be addressed for merit publication.

**Specific Comments:**

(1) Sect. 2.1, top-down estimation method:

My main concern lies on the top-down method. The present description in this section is not clear. The section states "the a posterior daily emissions were used as the a priori emissions of the next day, and the monthly top-down estimate of the NOx emissions was scaled from the average of the a posterior daily emissions of the last three days in the month". Do you mean the NOx emissions were calculated day by day for each month? In that case, there shall exists strong day-to-day variations in the top-down estimates, reflecting either true emission changes or uncertainties in satellite measurements and model results. It is then not proper to derive the monthly emission estimate based on only daily emissions in the last three days. This needs to be clarified in the manuscript and the daily emission variations if significant should be discussed.

(2) Page 4, Line 94 and Line 110:

"0.4 Tg N/yr" and "69.6 x 10$\hat{1}$3 molecules cm-2". Please also provide relative percentage numbers from the two studies, so that the magnitudes can be better understood.

(3) Page 7, Sect. 2.2 Model configuration:

What is domain 3 (D3) labelled in Figure 3? Is it used in this study?

(4) Page 8, Line 204-210:

Which year of data is used for the MEIC emission estimates?

[Figure]

(5) Page 9, Line 228-232:

Some previous studies (e.g., Lamsal et al., 2008; Liu et al. ACP 2018) suggested that the NO2 measurements obtained from the molybdenum-catalyzed conversion technique might be overestimated due to interference from other nitrogen species. Would this affect your results?

Lamsal, L. N., et al.: Ground-level nitrogen dioxide concentrations inferred from the satellite-borne Ozone Monitoring Instrument, J. Geophys. Res.- Atmos., 113, D16308, https://doi.org/10.1029/2007JD009235, 2008.

Liu, M., et al.: Spatiotemporal variability of NO2 and PM2.5 over Eastern China: observational and model analyses with a novel statistical method, Atmos. Chem. Phys., 18, 12933–12952, https://doi.org/10.5194/acp-18-12933-2018, 2018.

(6) Page 10, Line 258:

According to Figure 5 and 6, peaking ozone concentrations in YRD are also shown in the July month, and many previous studies have suggested more active ozone formation in summer. Some sentences are needed here to explain why this study focused on April and did not discuss July.

(7) Page 10, Sect. 3.1:

The spatial distribution of top-down vs. bottom-up NOx emission changes in YRD as shown in Figure S2 is an important finding of this study for explaining and supporting improvements in the top-down estimates. I suggest move Figure S2 to the main manuscript, e.g., combine with the present Figure 2.

(8) Page 13, Line 363 and 364:
Here "Fig. 5" should be "Fig. 4"

(9) Page 17, Line 465-469:

The decreases in the nitrate aerosol concentration in July with the top-down NOx emissions are interesting and worth further discussion. Reductions in NOx emissions would lead to increases in the nitrate aerosol concentration in other months (January, April, and July). Can you explain why the response in July is different from those in other months? Is it because the percentage reduction of top-down NOx emissions in July is much larger?

(10) Page 19, Line 538:
Should "Fig. 9b" here be "Fig. 9c"?

---

## Author Comment (AC2) · 21 Nov 2020

We thank very much for the valuable comments and suggestions from the reviewer, which help us improve our manuscript. The comments were carefully considered and revisions have been made in response to suggestions. Following are our point-by-point responses to the comments and corresponding revisions.

This manuscript has presented a top-down estimate of NOx emissions in the Yangtze River Delta (YRD) region and demonstrated that air quality modeling using the topdown NOx emissions could improve the simulations of ozone and secondary inorganic aerosol (SIA) over this region. A set of sensitivity simulations are conducted to better understand the formation of ozone and SIA under perturbed precursor emission conditions. This manuscript offers some new knowledge on the regional secondary pollution over YRD including an improved estimate of NOx emissions and predicted effectiveness of various emission controls on secondary pollution formation. This study is overall well conducted and analyzed. The manuscript is well written, and fits the scope of ACP. I think the following comments shall be addressed for merit publication.

Response and revisions:

We appreciate the reviewer's positive remarks.

Specific Comments:

1. Sect. 2.1, top-down estimation method: My main concern lies on the top-down method. The present description in this section is not clear. The section states "the a posteriori daily emissions were used as the a priori emissions of the next day, and the monthly top-down estimate of the NOx emissions was scaled from the average of the a posteriori daily emissions of the last three days in the month". Do you mean the NOx emissions were calculated day by day for each month? In that case, there shall exist strong day-to-day variations in the top-down estimates, reflecting either true emission changes or uncertainties in satellite measurements and model results. It is then not proper to derive the monthly emission estimate based on only daily emissions in the last three days. This needs to be clarified in the manuscript and the daily emission variations if significant should be discussed.

Response and revisions:

We thank the reviewer's important comment. Currently, the inverse model we applied in this work assumed that the daily emissions were similar (Zhao and Wang, 2009; Gu et al., 2014; Cooper et al., 2017). For example, the daily variation was expected to

be negligible over most regions of east China (Zhao and Wang, 2009). In our previous work (Yang et al., 2019), we evaluated the robustness of the method, by applying the "synthetic" TVCDs from air quality simulation based on a hypothetical "true" emission inventory, instead of those from satellite observation. We found that sufficient iteration times could result in a relatively constant emission estimate (the top-down estimate) close to the "true" emission input, implying the reliability of the inverse modeling method.

The assumption would bring some uncertainty as the daily variation of emissions did exist. Due mainly to the fair missed values of satellite detection, however, the daily variation could not be precisely captured by the top-down method, particularly at regional scale with relatively high horizontal resolution. Such method was designed for monthly mean of emissions. From a bottom-up perspective, the difference in NOX emissions between weekday and weekend was within 5% in the YRD region (Zhou et al., 2017), indicating an insignificant bias from ignoring the daily variation of emissions. We have added those descriptions in line 166 and lines 171-179 in the revised manuscript.

Reference: Zhao, C., and Wang, Y. X.: Assimilated inversion of NOx emissions over East Asia using OMI NO2 column measurements. Geophys. Res. Let., 2009, 36(L06805): 1-5.

Gu, D.S., Wang, Y.X., Smeltzer, C., Boersma, K.F.: Anthropogenic emissions of NOx over China: Reconciling the difference of inverse modeling results using GOME-2 and OMI measurements, J. Geophys. Res.: Atmosphere, 119, 7732-7740, 2014.

Cooper, M., Martin, R.V., Padmanabhan, A., Henze, D.K.: Comparing mass balance and adjoint methods for inverse modeling of nitrogen dioxide columns for global nitrogen oxide emissions, J. Geophys. Res.: Atmosphere, 122, 4718–4734, 2017.

Yang Y., Zhao Y., Zhang L., Lu Y.: Evaluating the methods and influencing factors of satellite-derived estimates of NOX emissions at regional scale: A case study for Yangtze River Delta, China. Atmos. Environ., 219, 1-12, 2019.
Zhou, Y.D., Zhao, Y.D., Mao, P., Zhang, Q., Zhang, J., Qiu, L.P., Yang, Y.: Development of a high-resolution emission inventory and its evaluation and application through air quality modeling for Jiangsu Province, China. Atmospheric Chemistry and Physics 17, 211-233, 2017.

2. Page 4, Line 94 and Line 110:"0.4 Tg N/yr" and "69.6 x 10ËĘ13 molecules cm-2". Please also provide relative percentage numbers from the two studies, so that the magnitudes can be better understood.

Response and revisions:

We thank the reviewer's comment. The relative percentage number for 0.4 Tg N/yr was 5.8% (Gu et al., 2014) and provided in line 96 in the revised manuscript, while that for 9.6 x 10ËĘ13 molecules cm-2 was unavailable in the original paper (Jena et al., 2014).

3. Page 7, Sect. 2.2 Model configuration: What is domain 3 (D3) labelled in Figure 1? Is it used in this study?

Response and revisions:

We thank the reviewer's reminder. The domain 3 (D3) in the original figure is not used in this study and thus removed in the revised Figure 1.

4. Page 8, Line 204-210: Which year of data is used for the MEIC emission estimates?

Response and revisions:

We thank the reviewer's reminder. The MEIC emission data for 2015 were used in this study and we have added the information in lines 213 in the revised manuscript.

5. Page 9, Line 228-232: Some previous studies (e.g., Lamsal et al., 2008; Liu et al. ACP 2018) suggested that the NO2 measurements obtained from the molybdenum-catalyzed conversion technique might be overestimated due to interference from other nitrogen species. Would this affect your results?

Lamsal, L. N., et al.: Ground-level nitrogen dioxide concentrations inferred from the satellite-borne Ozone Monitoring Instrument, J. Geophys. Res.- Atmos., 113, D16308, https://doi.org/10.1029/2007JD009235, 2008.

Liu, M., et al.: Spatiotemporal variability of NO2 and PM2.5 over Eastern China: observational and model analyses with a novel statistical method, Atmos. Chem. Phys., 18, 12933–12952, https://doi.org/10.5194/acp-18-12933-2018, 2018.

Response and revisions:

We thank the reviewer's very valuable comment. We agree with the reviewer that the NO2 concentration could be overestimated with the molybdenum-catalyzed conversion technique, while the effect of such overestimation on our results is expected to be limited. On one hand, the top-down estimates of NOX emissions were derived from satellite observation instead of ground observation. The observed ground NO2 concentrations were only used to evaluate the model performance with the bottom-up and top-down estimates of NOX emissions. On the other hand, as shown in Figure 4 in the revised manuscript, the simulation of NO2 concentration with the top-down estimates were improved by 30%-100% (indicated by the NMBs) compared to that with the bottom-up emission data, substantially larger than the common overestimation in NO2 observations with the measure around 15%. Therefore, the overestimation in ground-level NO2 concentrations could hardly change the basic judgment of this study that application of top-down estimates in NOX emissions would improve the model performance of NO2 concentration in the YRD region.

6. Page 10, Line 258: According to Figure 5 and 6, peaking ozone concentrations in YRD are also shown in the July month, and many previous studies have suggested more active ozone formation in summer. Some sentences are needed here to explain why this study focused on April and did not discuss July.

Response and revisions:

We thank the reviewer's comment. In the YRD region, on one hand, the peaking time of O3 concentrations has gradually moved forward from summer to late spring. In this work, for example, the mean observed O3 concentration of YRD in April was 72.5 $\mu$g/m3, even larger than that (71.9 $\mu$g/m3) in July. On the other hand, the model performance of O3 in this work was better for April than that for July (Fig. 6 in the revised manuscript). Therefore, we selected April to explore the sensitivity of O3 formation to precursor emissions. The corresponding revision was shown in lines 264-269 in the revised manuscript.

7. Page 10, Sect. 3.1: The spatial distribution of top-down vs. bottom-up NOx emission changes in YRD as shown in Figure S2 is an important finding of this study for explaining and supporting improvements in the top-down estimates. I suggest move Figure S2 to the main manuscript, e.g., combine with the present Figure 2.

Response and revisions:

We thank the reviewer's comment. We replot Figure S2 in the original submission and improve the figure quality. We move the figure to the main manuscript (Figure 3 in the revised manuscript).

8. Page 13, Line 363 and 364: Here "Fig. 5" should be "Fig. 4"

Response and revisions:

We are sorry for the error and thank the reviewer's reminder. "Fig. 5" should be "Fig. 4" in the original submission. As we add a figure in the revised manuscript (see our response to Question 7), now it should be Fig. 5 again. Hopefully this explanation is not confusing.

9. Page 17, Line 465-469: The decreases in the nitrate aerosol concentration in July with the top-down NOx emissions are interesting and worth further discussion. Reductions in NOx emissions would lead to increases in the nitrate aerosol concentration in other months (January, April, and July). Can you explain why the response in July is different from those in other months? Is it because the percentage reduction of top-down NOx emissions in July is much larger?

Response and revisions:

We thank the reviewer's important comment. It could result from two factors. First, the reduction of top-down NOx emissions in July was much larger, as suggested by the reviewer. Second, the VOC-limit mechanism in O3 formation was found weaker in summer than winter (see Fig. 7e and Fig. 7g), resulting in less O3 formation and thereby nitrate aerosol through oxidation. The corresponding revision was shown in lines 491-496 in the revised manuscript.

10. Page 19, Line 538: Should "Fig. 9b" here be "Fig. 9c"?

Response and revisions:

We are sorry for the error and thank the reviewer's reminder. "Fig. 9b" is now corrected to "Fig. 10c" (we add a new figure in the revised manuscript).

---

## Author Response (AR1)

**Main revisions and response to reviewers' comments**

Journal: Atmospheric Chemistry and Physics

Manuscript No.: acp-2020-751

Title: Improvement from the satellite-derived NOx emissions on air quality modeling and its effect on ozone and secondary inorganic aerosol formation in Yangtze River Delta, China

Author: Yang Yang, Yu Zhao, Lei Zhang, Jie Zhang, Xin Huang, Xuefen Zhao, Yan Zhang, Mengxiao Xi and Yi Lu

We thank very much for the valuable comments and suggestions from the two reviewers, which help us improve our manuscript. The comments were carefully considered and revisions have been made in response to suggestions. Following are our point-by-point responses to the comments and corresponding revisions.

**Reviewer #1:**

0. The authors developed a "top-down" methodology based on the inversed chemistry transport modeling and satellite data to estimate the NOx emissions for four seasons in YRD region in 2016. The results show that the improved NO2, O3, and SNA simulation results can be achieved with top-down estimates comparing to current bottom-up estimates. Further sensitivity study of O3 formation indicates the effectiveness of controlling VOCs emissions on O3 pollution abatement for PRD region and reducing NH3 emissions was crucial to alleviate SNA pollution of YRD in winter. The manuscript was generally well written, the research presented is innovative and the results can guide the policymaking. I recommend this paper to be published in ACP after some comments have been addressed.

**Response and revisions:**

We appreciate the reviewer's positive remarks.

1. Please revised the introduction part thoroughly to improve the narrative logic, the current version is a little hard to follow and some statements need to be summarized.

**Response and revisions:**

We thank the reviewer's comment. We thoroughly checked and revised the introduction. The section was better paragraphed to make the narrative logic clear. Some distracting sentences were deleted, and some summarizing phrases were added in corresponding positions. Now the main contents of introduction include 1) the importance of NOX emission inventory and its bottom-up development method; 2) the top-down method; 3) the more application of the top-down method at the global/national scale compared to the regional scale (limitation); 4) the evaluation of the top-down emission estimates and limitation; and 5) summary of main tasks of this work.

2. Line 259-265: The description of Table S3 does not agree with Table S3 shown in the Supplement file. And please clarify the meaning of "-" in Table S3, preferably with a footnote.

**Response and revisions:**

We thank the reviewer's reminder and we are sorry for the error. The description for Table S3 was corrected in the revised manuscript (Lines 271-277) and now the statement agrees with Table S3. The meaning of "-" in the original table was that the emissions was not changed, and "No change" in the revised table is applied instead of "-" to avoid the confusion.

3. Line 386-389: Why did the authors only perform an extra simulation of exploring the influence of BVOCs emissions with top-down estimate instead of with both top-down and bottom-up estimates to prove that a better  $O_3$  simulation can be achieved based on top-down NOx estimates? Please clarify it.

**Response and revisions:**

We thank the reviewer's very valuable comment. The evaluation of emission inventory could be complicated with different species included. In this work, as shown in Figure 4c (Figure 3c in the original submission) in the revised manuscript, very clear improvement in NO2 simulation was found with the top-down NOX estimates for July, implying the improved emission estimation with the satellite constraint. The O3 simulation for July, however, was poorer when top-down estimate was applied (Figure 6c). We expected many other factors contributed to the uncertainty in O3 simulation, besides the NOX emission input. One possible factor could be the overestimation of BVOCs emissions. That's why we performed an extra case by reducing half of BVOCs emissions in the YRD region. Although the model performance was improved compared to the case without BVOCs reduction (Figure S3 in the revised supplement), still it was poorer than the case with bottom-up  $NO_X$ emission estimates applied (note the NMBs with bottom-up NOX emissions applied was very small at 1.1% in Figure 6c). This comparison thus suggested that the complicated mechanism for summer  $O_3$  formation was insufficiently considered in current model, and it is partly out of scope of current paper. We clarified this in lines 393-396 and lines 404-407 in the revised manuscript.

4. Line 409-413: Please add references after these two statements.

**Response and revisions:**

We thank the reviewer's comment. We add references (Wang et al., 2019 and Li, 2019) after the two statements.

Reference:

Wang, N., Lyu, X., Deng, X., Huang, X., Jiang, F., Ding, A.: Aggravating O3 pollution due to NOx emission control in eastern China, Sci. Total Environ., 677, 732-744, 2019.

Li, L.: Application of new generation natural source emission model in Yangtze River Delta and its influence on SOA and  $O_3$  (in Chinese), The 4th application technology seminar on air pollution source emission inventory in China, Nanjing, China, September 18-19, 2019.

5. Line 423-426: Please explain more to support the inference and can authors replot figure S2? The current one is blurring.

**Response and revisions:**

We thank the reviewer's comment. We explain more to support the inference in lines 440-445 in the revised manuscript: As east-central YRD was located in a VOC-limited region, the  $O_3$  concentrations of the region would be elevated along with the reduced NOX emissions (Wang et al., 2019). The comparison between Figure 7 and Figure S2 (original submission) thus reflects the negative effect of NOX control on  $O_3$  pollution alleviation in the region. We also replot Figure S2 and improve the figure quality in the revised supplement. We move to the figure to the main manuscript as Figure 3 (please see our response to Question 7 of Reviewer #2).

**Reference:**

Wang, N., Lyu, X., Deng, X., Huang, X., Jiang, F., Ding, A.: Aggravating O3 pollution due to NOx emission control in eastern China, Sci. Total Environ., 677, 732-744, 2019.

6. Line 427: I think changing SIA to SNA would be better to keep the consistency of the full text.

**Response and revisions:**

We thank the reviewer's comment and have changed SIA to SNA in the full text.

7. Line 451-453: Sha et al. (2019) reported that SO2 heterogeneous oxidation can largely improve the sulfate simulation results in Nanjing. Authors may incorporate the related mechanisms to perform the simulation, if possible, or at least mention this potential reason when discussing the factors influencing the accuracy of SNA simulation. References: Sha T, Ma X, Jia H, Tian R, Chang Y, Cao F, Zhang Y. Aerosol chemical component: Simulations with WRF-Chem and comparison with observations in Nanjing. Atmospheric Environment. 2019 Dec 1; 218: 116982.

**Response and revisions:**

We thank the reviewer's important comment. We agree with the author that the chemical mechanisms in the model could be important for model performance. We have added the reference (Sha et al., 2019) and the statement that  $SO_2$  heterogeneous oxidation can largely improve the sulfate simulation results in Nanjing in lines 472-475 in the revised manuscript.

**Reference:**

Sha, T., Ma, X., Jia, H., Tian, R., Chang, Y., Cao, F., Zhang, Y.: Aerosol chemical component: Simulations with WRF-Chem and comparison with observations in Nanjing, Atmos. Environ., 218 (116982), 1-14, 2019.

**Reviewer #2:**

This manuscript has presented a top-down estimate of NOx emissions in the Yangtze River Delta (YRD) region and demonstrated that air quality modeling using the top-down NOx emissions could improve the simulations of ozone and secondary inorganic aerosol (SIA) over this region. A set of sensitivity simulations are conducted to better understand the formation of ozone and SIA under perturbed precursor emission conditions. This manuscript offers some new knowledge on the regional secondary pollution over YRD including an improved estimate of NOx emissions and predicted effectiveness of various emission controls on secondary pollution formation. This study is overall well conducted and analyzed. The manuscript is well written, and fits the scope of ACP. I think the following comments shall be addressed for merit publication.

**Response and revisions:**

We appreciate the reviewer's positive remarks.

**Specific Comments:**

1. Sect. 2.1, top-down estimation method:

My main concern lies on the top-down method. The present description in this section is not clear. The section states "the a posterior daily emissions were used as the a priori emissions of the next day, and the monthly top-down estimate of the NOx emissions was scaled from the average of the a posterior daily emissions of the last three days in the month". Do you mean the NOx emissions were calculated day by day for each month? In that case, there shall exist strong day-to-day variations in the top-down estimates, reflecting either true emission changes or uncertainties in satellite measurements and model results. It is then not proper to derive the monthly emission estimate based on only daily emissions in the last three days. This needs to be clarified in the manuscript and the daily emission variations if significant should be discussed.

**Response and revisions:**

We thank the reviewer's important comment. Currently, the inverse model we applied in this work assumed that the daily emissions were similar (Zhao and Wang, 2009; Gu et al., 2014; Cooper et al., 2017). For example, the daily variation was expected to be negligible over most regions of east China (Zhao and Wang, 2009). In our previous work (Yang et al., 2019), we evaluated the robustness of the method, by applying the "synthetic" TVCDs from air quality simulation based on a hypothetical "true" emission inventory, instead of those from satellite observation. We found that sufficient iteration times could result in a relatively constant emission estimate (the top-down estimate) close to the "true" emission input, implying the reliability of the inverse modeling method.

The assumption would bring some uncertainty as the daily variation of emissions did exist. Due mainly to the fair missed values of satellite detection, however, the daily variation could not be precisely captured by the top-down method, particularly at regional scale with relatively high horizontal resolution. Such method was designed for monthly mean of emissions. From a bottom-up perspective, the difference in  $NO_X$  emissions between weekday and weekend was within 5% in the YRD region (Zhou et al., 2017), indicating an insignificant bias from ignoring the daily variation of emissions. We have added those descriptions in line 166 and lines 171-179 in the revised manuscript.

**Response and revisions:**

We are sorry for the error and thank the reviewer's reminder. "Fig. 9b" is now corrected to "Fig. 10c" (we add a new figure in the revised manuscript).

| 1              |                                                                                                                                                                                         |
|----------------|-----------------------------------------------------------------------------------------------------------------------------------------------------------------------------------------|

[revised manuscript text omitted]

|--------|-------------------------------------------------------------------------------|
| \
| ۱
|        | top-down estimates resulting from the                                         |
|        | uncertainties of the inversed method                                          |
|        | and satellite observation (Cooper et al                                       |
|        | 2017; Dilig et al., 2017; Liu et al.,
2019: Yang et al., 2019: b) and they |
|        | could further influence the reliability                                       |
|        | of AQM and the rationality of control                                         |
|        |                                                                        |

|----------------|--|
| 别除的内容:  |  |
| 别除的内容:  |  |
| 削除的内容:  |  |

ł

ł

ł

ł

4

between the NO2 TVCDs from OMI observation and AQM based on the top-down

 $NO_X$  emission estimation was -30.8+69.6×1013 molecules cm-2 in winter in India

(Jena et al., 2014). The linear correlation coefficient  $(R^2)$  between OMI and AQM

123

124

125

126

with the top-down emission estimates could reach 0.84 in Europe (Visser et al., 2019).

Compared to the satellite observation with relatively large uncertainty (Yang et al., 173 174 2019b; Liu et al., 2019), surface concentrations that better represent the effect of air pollution on human health and the ecosystems were less applied in the evaluation of 175 176 the top-down estimates of NOX emissions. Limited assessments were conducted at the 177 national scale. For example, Liu et al. (2018) found that the normalized mean error (NME) between the observed and simulated NO2 concentrations based on the 178 top-down estimate of NOX emissions could reach 32% in China at the resolution of 179  $0.25^{\circ}\times 0.25^{\circ}$ . Besides NO2, the estimation of NOx emissions also play an important 180 and complicated role on secondary air pollutant simulation including  $O_3$  and SNA, 181 and the response of secondary pollution to the primary emissions was commonly 182 183 nonlinear. The simulated O3 concentrations in Shanghai (the most developed city in eastern China) could increase over 20% with a 60% reduction in NOX emissions in 184 summer 2016, implying a clear "VOC-limit" pattern for the  $O_3$  formation in the mega 185 186 city (Wang et al., 2019). For the response of SNA to NOX emissions, the NH4+ and SO42- concentrations at an urban site in another mega city Nanjing in eastern China 187 were simulated to increase 1.9% and 2.8% with a 40% abatement of NOX emissions 188 189 in autumn 2014, respectively, due to the weakened competition of SNA formation against SO2 (Zhao et al., 2020). To our knowledge, however, the relatively new 190 191 information from the inversed modeling of NOX emissions has not been sufficiently 192 incorporated into the SNA and O3 analyses with AQM in China. 193 Located in eastern China, the Yangtze River Delta (YRD) region including the 194 city of Shanghai and the provinces of Anhui, Jiangsu and Zhejiang is one of the most developed and heavy-polluted regions in the country. The air quality for most cities in 195 196 YRD failed to meet National Ambient Air Quality Standard (NAAQS) Class II in 2016 (MEPPRC, 2017). NOX emissions made great contributions to the severe air 197

2016 (MEPPRC, 2017). NOX emissions made great contributions to the severe air pollution in the region. Based on an offline-sampling and measurement study, for example, the annual average of the NO3- mass fraction to the total PM2.5 reached 19% in Shanghai in 2014, and it was significantly elevated in the pollution event periods (Ming et al., 2017). In this study, we chose the YRD to estimate the NOX emissions

with the inversed method and to explore their influence on the air quality modeling.

resolutions

[revised manuscript text omitted]
 O3 concentration in April was 72.5 µg/m3, slightly higher than 343 that in July (71.9  $\mu$ g/m3). In addition, the model performance of O3 was better for 344 April than that for July in this work (see details in Section 3.2). Therefore, we selected 345 April to explore the sensitivity analysis of O3 formation in the region. As summarized 346 in Table S3 in the supplement, eight cases were set besides the base scenario with the 347 top-down NOX estimates for April 2016, Cases 1 and 6 reduced only the NOX 348 emissions by 30% and 60%, and Cases 2 and 7 reduced only the VOCS emissions by 349 30% and 60%, respectively. To explore the co-effect of VOCs and NOX emission 350 controls on O3 concentrations, cases with different reduction rates of VOCs and NOX 351 emissions were designed. The emissions of  $NO_X$  and VOCs in Case 4 were decreased 352 by 30% and 60%, and in Case 5 by 60% and 30%, respectively. Both NOX and VOCs 353 emissions were reduced 30% and 60% in Cases 3 and 8, respectively. 354 355 The response of SNA concentrations to the changes in precursor emissions was influenced by various factors including the abundance of NH3, atmospheric oxidation, 356 and the chemical regime of  $O_3$  formation (Wang et al., 2013; Cheng et al., 2016; Zhao 357 358 et al., 2020). To explore the sensitivity of SNA formation to its precursor emissions, four cases were set besides the base scenario for January 2016, the month with the 359 largest observed SNA concentrations. As shown in Table S4 in the supplement, the 360

|---|--------|--|
|---|--------|--|

363 Case 12.

361

362

[revised manuscript text omitted]
 NO2 concentration and suppressed the formation of NO3, while the enhanced O3 from the reduced NOX emissions promoted it (Cai et al., 2017; Huang et al., 2020). In 573 summer, the former dominated the process with the most evident improvement in NO2 574 simulation (Figure 4), thus the reduced NO3- concentrations that were closer to 575 576 observation were simulated for all the cities.

The simulations with both top-down and bottom-up estimates of  $NO_X$  emissions underestimated the  $NH_4^+$  concentrations for most cases, and such underestimation was slightly corrected with the application of the top-down estimates except for summer. 删除的内容: ure S2 删除的内容: resulted in 删除的内容: more 删除的内容: , 带格式的: 下标 删除的内容: abatement 删除的内容: control in the VOC-limited regions 删除的内容: Since east-central YRD was located in VOC-limited region, the O3 concentrations of the region increased due to reduction of NOX emissions (Wang et al., 2019).

[revised manuscript text omitted]
                                                                                                                                                                                                                                                                                                                                   | 600 | simulated and observed $\mathrm{NH_4}^{\scriptscriptstyle +}.$ The difference between the simulated $\mathrm{SO_4}^{2\text{-}}$ with the |                      |                  |
| 602 implying a limited benefit of improved NO X emissions on SO 4 2- modeling. Besides
603 emission data, the chemical mechanisms included in the model should be important
604 for the model performance. For example, adding SO 2 heterogeneous oxidation in the
605 model could Jargely improve the sulfate simulation in Nanjing (Sha et al., 2019)
606 Figure 8 shows the differences in the spatial distribution of SNA concentrations
607 simulated with the bottom-up and top-down estimates of NO X emissions by month. In
608 most of the region, the differences of NO 3 - concentrations were larger than those of
609 NH 4 + and SO 4 2- for all seasons, and they were mainly controlled by the changed
610 ambient NO 2 or O 3 level. The difference in spatial pattern of NO 3 - was similar to that
611 of O 3 for January, and the larger growth attributed to the application of the top-down
612 estimates was found in northern Anhui and eastern Zhejiang (Fig. 8a ). The result
613 implies that the change in NO 3 - concentration in winter could result partly from the
614 improved O 3 simulation, i.e., the elevated O 3 was an important reason for the
615 enhanced the formation of SNA in winter (Huang et al., 2020). Similarly, the
616 increased NO 3 - was found for more than half of the YRD region in April, along with
617 the growth of O 3 concentrations (Fig. 8d ). For July, however, the difference in spatial
618 pattern of NO 3 - (Fig. 8g ) was similar with NO 2 (Fig. 5g ), and the larger reduction
619 attributed to the application of the top-down estimates was found in northern YRD.
620 The result suggests that the declining NO x emissions and thereby NO 2 concentration
621 dominated the reduced NO 3 - formation in summer. It was mainly because the
622 reduction of top-down NO x emission, estimate from the bottom-up emission inventory,
623 was much larger for July compared to spring or autumn (Fig 2). In addition, the                                                                                                                                     | 601 | bottom-up and top-down $\ensuremath{\text{NO}}_X$ emission estimates were small for most seasons,                                        |                      |                  |
|  <li>emission data, the chemical mechanisms included in the model should be important</li> <li>for the model performance. For example, adding SO2 heterogeneous oxidation in the</li> <li>model could Jargely improve the sulfate simulation in Nanjing (Sha et al., 2019)</li> <li>Figure & shows the differences in the spatial distribution of SNA concentrations</li> <li>simulated with the bottom-up and top-down estimates of NOx emissions by month. In</li> <li>most of the region, the differences of NO3- concentrations were larger than those of</li> <li>NH4+ and SO42- for all seasons, and they were mainly controlled by the changed</li> <li>ambient NO2 or O3 level. The difference in spatial pattern of NO3- was similar to that</li> <li>of O3 for January, and the larger growth attributed to the application of the top-down</li> <li>estimates was found in northern Anhui and eastern Zhejiang (Fig. 8a). The result</li> <li>implies that the change in NO3- concentration in winter could result partly from the</li> <li>improved O3 simulation, i.e., the elevated O3 was an important reason for the</li> <li>enhanced the formation of SNA in winter (Huang et al., 2020). Similarly, the</li> <li>increased NO3- was found for more than half of the YRD region in April, along with</li> <li>the growth of O3 concentrations (Fig. 8d). For July, however, the difference in spatial</li> <li>pattern of NO3- (Fig. 8g) was similar with NO2 (Fig. 5g), and the larger reduction</li> <li>attributed to the application of the top-down estimates was found in northern YRD.</li> <li>The result suggests that the declining NOx emissions and thereby NO2 concentration</li> <li>dominated the reduced NO3- formation in summer. It was mainly because the</li> <li>reduction of top-down, NOx emission, estimate from the bottom-up emission inventory.</li> <li>was much larger for July compared to spring or autumn (Fig 2). In addition, the</li>                                                                                                                                                                                                                                                  | 602 | implying a limited benefit of improved $\mathrm{NO}_X$ emissions on $\mathrm{SO_4}^{2\text{-}}$ modeling. Besides                        |                      |                  |

[revised manuscript text omitted]
</li>                                                                                                                                                                                                                                                                                                                                                                                                                                                                                                                                                                                                                                                                                                                                                                                                                                                                                                                                                                                                                                                                                                                                                                                                                                                                                                                                                                                                                                                                                                                                                                                                                                                                                                                                                                                                                                                                                                                                                                                                                                                                                                                                                                                                                                                                                | 620 | The result suggests that the declining $\mathrm{NO}_{\mathrm{X}}$ emissions and thereby $\mathrm{NO}_2$ concentration                    |                      | 市柏
删除         |
| 622       reduction of top-down NOx emission estimate from the bottom-up emission inventory,         623       was much larger for July compared to spring or autumn (Fig 2). In addition, the                                                                                                                                                                                                                                                                                                                                                                                                                                                                                                                                                                                                                                                                                                                                                                                                                                                                                                                                                                                                                                                                                                                                                                                                                                                                                                                                                                                                                                                                                                                                                                                                                                                                                                                                                                                                                                                                                                                                                                                                                                                                                                                                                                                                                                                                                                                                                                                                           | 621 | dominated the reduced $NO_3^-$ formation in summer. It was mainly because the                                                            |                      | 删除               |
| 623 was much larger for July compared to spring or autumn (Fig 2). In addition, the                                                                                                                                                                                                                                                                                                                                                                                                                                                                                                                                                                                                                                                                                                                                                                                                                                                                                                                                                                                                                                                                                                                                                                                                                                                                                                                                                                                                                                                                                                                                                                                                                                                                                                                                                                                                                                                                                                                                                                                                                                                                                                                                                                                                                                                                                                                                                                                                                                                                                                                      | 622 | reduction of top-down NO x emission estimate from the bottom-up emission inventory,                                           |                      | 删除               |
|                                                                                                                                                                                                                                                                                                                                                                                                                                                                                                                                                                                                                                                                                                                                                                                                                                                                                                                                                                                                                                                                                                                                                                                                                                                                                                                                                                                                                                                                                                                                                                                                                                                                                                                                                                                                                                                                                                                                                                                                                                                                                                                                                                                                                                                                                                                                                                                                                                                                                                                                                                                                          | 623 | was much larger for July compared to spring or autumn (Fig 2). In addition, the                                                          |                      | 删除               |

| 删除的内容:     | It was partly because of |
|------------|--------------------------|
| incomplete |                          |

|--------------------------------|

Fig. 7e and Fig. 7g), resulting in less O3 formation and thereby nitrate aerosol through 644 oxidation. In October, the growth in NO3- concentrations was found again in most 645 646 YRD when the top-down estimates were applied (Fig. §i). The growth in the north resulted mainly from the increased O3 level, while that in the south was associated 647 with the increased NO2. The differences in spatial patterns of simulated NH4+ 648 concentrations were similar to those of NO3- 
[revised manuscript text omitted]

|---|------------------|

|----|------------------|

that of  $SO_4^{2-}$ . As NH3 reacted with SO2 prior to NOX, NH4NO3 was assumed easier to decompose than (NH4)2SO4 when NH3 emissions were reduced. The growth of NO3- concentrations was found for Case 10 (SO2 control only), since the free NH3 from the reduced SO2 emissions could react with NOX in the NH3-poor condition. Similarly, the SO42- concentrations increased for Case 9 (NOX control only), as the elevated O3 attributed to the reduction of NOX emissions promoted the SO42- formation.

757

**4. Summary**

From a "top-down" perspective, we have estimated the monthly NOX emissions 758 for the YRD region in 2016, based on the nonlinear inversed modeling and NO2 759 TVCDs from POMINO, and the bottom-up and top-down estimates of NOx emissions 760 were evaluated with AQM and ground NO2 observation. Due to insufficient 761 consideration of improved controls on power and industrial sources, the NOX 762 emissions were probably overestimated in current bottom-up inventory (MEIC), 763 resulting in significantly higher simulated NO2 concentrations than the observation. 764 765 The simulated  $NO_2$  concentrations with the top-down estimates were closer to the 766 observation for all the four seasons, suggesting the improved emission estimation with satellite constraint. Improved O3 and SNA simulations with the top-down NOX 767 768 estimates for most months indicate the importance role of precursor emission 769 estimation on secondary pollution modeling for the region. Through the sensitivity 770 analysis of  $O_3$  formation, the mean  $O_3$  concentrations were found to decrease for most YRD when only VOCs emissions were reduced or the reduced rate of VOCs was 771 twice of  $NO_X$ , and the result indicates the effectiveness of controlling VOCs 772 773 emissions on O3 pollution abatement for the region. For part of southern Zhejiang, however, the  $O_3$  concentrations were simulated to decline with the reduced  $NO_X$ 774 775 emissions, implying the shifting from VOC-limited to NOX-limited region. Compared to reducing NOX or SO2 only, larger reduction in SNA concentrations was found when 776 30% of emissions were cut for NH3 or all the three precursors (NO2, NH3 and SO2). 777 The result suggests that reducing NH3 
[revised manuscript text omitted]